

**Insights on Atmospheric Oxidation Processes by Performing Factor Analyses on**
**Sub-ranges of Mass Spectra**
Yanjun Zhang[1], Otso Peräkylä[1], Chao Yan[1], Liine Heikkinen[1], Mikko Äijälä[1], Kaspar R. Daellenbach[1],
Qiaozhi Zha[1], Matthieu Riva[1,2], Olga Garmash[1], Heikki Junninen[1,3], Pentti Paatero[1], Douglas Worsnop[1,4], and
Mikael Ehn[1]
[1] Institute for Atmospheric and Earth System Research / Physics, Faculty of Science, University of Helsinki,
Helsinki, 00014, Finland
[2] Univ Lyon, Université Claude Bernard Lyon 1, CNRS, IRCELYON, F-69626, Villeurbanne, France
[3] Institute of Physics, University of Tartu, Tartu, 50090, Estonia
[4] Aerodyne Research, Inc., Billerica, MA 01821, USA
Corresponding author: yanjun.zhang@helsinki.fi
**Abstract**
With the recent developments in mass spectrometry, combined with the strengths of factor analysis
techniques, our understanding of atmospheric oxidation chemistry has improved significantly. The
typical approach for using techniques like positive matrix factorization (PMF) is to input all measured
data for the factorization in order to separate contributions from different sources and/or processes to
the total measured signal. However, while this is a valid approach for assigning the total signal to
factors, we have identified several cases where useful information can be lost if solely using this
approach. For example, gaseous molecules emitted from the same source can show different temporal
behaviors due differing loss terms, like condensation at different rates due to different molecular
masses. This conflicts with one of PMF's basic assumptions of constant factor profiles. In addition,
some ranges of a mass spectrum may contain useful information, despite contributing only minimal
fraction to the total signal, in which case they are unlikely to have a significant impact on the
factorization result. Finally, certain mass ranges may contain molecules formed via pathways not
available to molecules in other mass ranges, e.g. dimeric species versus monomeric species. In this
study, we attempted to address these challenges by dividing mass spectra into sub-ranges and
applying the newly developed binPMF method to these ranges separately. We utilized a dataset from
a chemical ionization atmospheric pressure interface time-of-flight (CI-APi-TOF) mass spectrometer
as an example. We compare the results from these three different ranges, each corresponding to
molecules of different volatilities, with binPMF results from the combined range. Separate analysis
showed clear benefits in dividing factors for molecules of different volatilities more accurately, in



resolving different chemical processes from different ranges, and in giving a chance for high-
molecular-weight molecules with low signal intensities to be used to distinguish dimeric species with
different formation pathways. In addition, daytime dimer formation (diurnal peak around noon) was
identified, which may contribute to NPF in Hyytiälä. Also, dimers from $NO_3$ oxidation were separated
by the sub-range binPMF, which would not be identified otherwise. We recommend PMF users to try
running their analyses on selected sub-ranges in order to further explore their datasets.

**1 Introduction**
Huge amounts of volatile organic compounds (VOC) are emitted to the atmosphere every year
(Guenther et al., 1995;Lamarque et al., 2010), which play a significant role in atmospheric chemistry
and affect the oxidative ability of the atmosphere. The oxidation products of VOC can contribute to
the formation and growth of secondary organic aerosols (Kulmala et al., 2013;Ehn et al., 2014;Kirkby
et al., 2016;Troestl et al., 2016), affecting air quality, human health, and climate radiative forcing
(Pope III et al., 2009;Stocker et al., 2013;Zhang et al., 2016;Shiraiwa et al., 2017). Thanks to the
advancement in mass spectrometric applications, like the aerosol mass spectrometer (AMS)
(Canagaratna et al., 2007) and chemical ionization mass spectrometry (CIMS) (Bertram et al.,
2011;Jokinen et al., 2012;Lee et al., 2014) our capability to detect these oxidized products, as well as
our understanding of the complicated atmospheric oxidation pathways in which they take part, have
been greatly enhanced.
Monoterpenes ($C_{10}H_{16}$), one common group of VOC emitted in forested areas, have been shown to
be a large source of atmospheric secondary organic aerosol (SOA). The oxidation of monoterpenes
produces a wealth of different oxidation products (Oxygenated VOC, OVOC), including highly
oxygenated organic molecules (HOM) with molar yields in the range of a few percent, depending on
the specific monoterpene and oxidant (Ehn et al., 2014;Bianchi et al., 2019). Bianchi et al. (2019)
summarized that HOM can be either Extremely Low Volatility Organic Compounds (ELVOC), Low
Volatility Organic Compounds (LVOC), or Semi-volatile Organic Compounds (SVOC)
(classifications by Donahue et al. 2012), depending on their exact structures. For less oxygenated
products, the majority are likely to fall into the SVOC or the Intermediate VOC (IVOC) range. The
volatility of the OVOC will determine their dynamics, including their ability to contribute to the
formation of SOA and new particles (Bianchi et al., 2019;Buchholz et al., 2019).
The recent developments of CIMS techniques has allowed researchers to observe unprecedented
numbers of OVOC, in real-time (Riva et al., 2019). This ability to measure thousands of compounds
is a great benefit, but also a large challenge for the data analyst. For this reason, factor analytical
techniques have often been applied to reduce the complexity of the data by finding co-varying signals



that can be grouped into common factors (Huang et al., 1999). For aerosol and gas-phase mass
spectrometry, positive matrix factorization, PMF (Paatero and Tapper, 1994;Zhang et al., 2011) has
been the most utilized tool. The factors have then been attributed to sources (e.g. biomass burning
organic aerosol) or processes (e.g. monoterpene ozonolysis) depending on the application and ability
to identify spectral signatures (Yan et al., 2016;Zhang et al., 2017). In the vast majority of these PMF
applications to mass spectra, the mass range of ions has been maximized in order to provide as much
input as possible for the algorithm. This approach was certainly motivated in early application of
PMF on e.g. offline filters, with chemical information of metals, water-soluble ions, and organic and
elemental carbon (OC and EC), where the number of variables is counted in tens, and the number of
samples in tens or hundreds (Zhang et al., 2017). However, with gas-phase CIMS, we often have up
to a thousand variables, with hundreds or even thousands of samples, meaning that the amount of data
itself is unlikely to be a limitation for PMF calculation. In this work, we aimed to explore potential
benefits of dividing the spectra into sub-ranges before applying factorization analysis.
An inherent requirement of factorization approaches is that the factor profiles, in this case the relative
abundancies of ions in the mass spectra, of each factor stay nearly constant. Due to the complexity
and number of atmospheric processes affecting the formation, transformation, and loss of VOC,
OVOC and aerosol, this often does not hold, and is one of the main limitations of factorization
approaches. Given the different volatilities of OVOC, it may even be expected that molecules from
the same source may have very different loss time scales, which may affect the factor analysis.
Volatility issue has been studied and reported for AMS data, with different volatilities of various OA
types (Huffman et al., 2009;Crippa et al., 2014;Paciga et al., 2016;Äijälä et al., 2017). Semi-volatile
oxygenated organic aerosol (SV-OOA) and Low-volatility oxygenated organic aerosol (LV-OOA)
can both be mainly produced from biogenic sources, but get separated based on different volatilities
by PMF (El Haddad et al., 2013). Sekimoto et al. (2018) found that the two profiles resolved with
VOC emitted from biomass burning had different estimated volatilities. As the volatility of a molecule
is linked to its molecular mass (Peräkylä et al., 2019), it may be beneficial to apply PMF separately
to mass ranges where one can expect the loss processes to be similar, thereby resulting in more
constant factor profiles. In this way, distinct sources are hopefully separated by PMF, with minimized
influence of differing volatilities from one source.
The number of PMF or other factorization studies utilizing CIMS data remains very limited.
"Traditional" PMF analyses have so far, to our knowledge, only been applied to nitrate-based
chemical ionization atmospheric pressure interface time-of-flight (CI-APi-TOF) data (Yan et al.,
2016;Massoli et al., 2018). One study has also utilized non-negative matrix factorization (NNMF) to
look at diurnal trends of Iodide ToF-CIMS data (Lee et al., 2018). The lack of more studies utilizing



PMF, or other factorization techniques, on CIMS data is likely partly due to the complexity of the
data, with multiple overlapping ions hampering HR peak fitting (Zhang et al., 2019). In addition,
variable factor profiles may hamper PMF's ability to correctly separate the factors. The two CI-APi-
TOF studies utilized the nearly the entire measured spectrum (from around 200 Th to around 600 Th),
either in unit mass resolution (UMR) or high resolution (HR) peak fitting data (Yan et al.,
2016;Massoli et al., 2018). Massoli et al. (2018) estimated the volatility of the molecules they detected,
finding that all the six extracted factors had notable contributions from IVOC, SVOC and (E)LVOC.
These compound groups will have clearly different loss mechanisms, and thereby loss rates,
suggesting that variation in factor profiles is inevitable, even if the source was identical for all
molecules in the factor. We hypothesize that this effect further hampers the correct factorization, and
further that this effect can be reduced by dividing the spectra into separate ranges, with each sub-
range containing molecules with roughly similar loss mechanisms and rates.
As an additional motivation to separate different ranges from the mass spectrum, it is not only the
loss mechanisms, but also the formation pathways that may differ. For example, atmospheric
oxidation chemistry of organics is, to a large extent, the chemistry of peroxy radicals ($RO_2$) (Orlando
and Tyndall, 2012).These $RO_2$ are initiated by VOC reacting with oxidants like ozone, or the hydroxyl
(OH) or nitrate ($NO_3$) radicals, while their termination occurs mainly by bimolecular reactions with
NO, $HO_2$ and/or other $RO_2$. Some product molecules can be formed from many of the three
termination pathways, while for example ROOR "dimers" can only be formed from $RO_2+RO_2$
reactions (Berndt et al., 2018a;Berndt et al., 2018b). This also means that there can be six different
pathways to form dimers from the same precursors VOC, by combining $RO_2$ formed from the same
or different oxidants. As an example of the latter, an ROOR dimer can contain one moiety produced
from ozone oxidation and another moiety from $NO_3$ oxidation (Yan et al., 2016). Thus, their
concentration is dependent on both the precursor VOC concentration, and the concentrations of both
oxidants. Such a molecule will not have a direct equivalent in any of the monomer products,
dependent on only one oxidant, which again may complicate the separation of such factors by PMF,
if the entire spectrum is analyzed once. However, if separating the monomer and dimer products
before PMF analysis, separation of different formation pathways can potentially become simpler.
Recently, we proposed a new PMF approach, binPMF, to simplify the analysis of mass spectral data
(Zhang et al., 2019). This method divides the mass spectrum into narrow bins, typically some tens of
bins per integer mass, depending on the mass resolving power of the instrument, before performing
PMF analyses. In this way, binPMF does not require any time-consuming, and potentially subjective
high resolution peak fitting, and can thus be utilized for data exploration at a very early stage of data
analysis. Data preparation is nearly as simple as in the case of UMR analysis, yet it utilizes much



more spectral information as it does not sum up signal over all ions at each integer mass. In addition
to saving time and effort in data analysis, the results are less sensitive to mass calibration fluctuations.
Finally, the binning also greatly increases the number of input variables, which has the advantage that
factor analysis with smaller mass ranges becomes more feasible, as more meaningful variation is
supplied to the algorithm.
We designed this study to explore the benefits of separate analysis of different mass ranges from mass
spectra. We used a previously published ambient dataset measured by a CI-APi-TOF, and conducted
binPMF analysis with three different mass ranges, roughly corresponding to different volatility ranges.
We compared the results from the sub-range analyses with each other and with results from binPMF
run on the combined ranges. We found that more meaningful factors are separated from our dataset
by utilizing the sub-ranges, and believe that this study will provide new perspectives for future studies
analyzing gas-phase CIMS data.

**2 Methodology**
The focus of this work is on retrieving new information from mass spectra by applying new analytical
approaches. Therefore, we chose a dataset that has been presented earlier, though without PMF
analysis, by Zha et al. (2018), and was also used in the first study describing the binPMF method
(Zhang et al., 2019). The measurements are described in more details below in section 2.1, while the
data analysis techniques used in this work are presented in section 2.2.
2.1 Measurements
2.1.1 Ambient site
The ambient measurements were conducted at the Station for Measuring Ecosystem–Atmosphere
Relations (SMEAR) II in Finland (Hari and Kulmala, 2005) as part of the Influence of Biosphere-
Atmosphere Interactions on the Reactive Nitrogen budget (IBAIRN) campaign (Zha et al, 2018).
Located in the boreal environment in Hyytiälä, SMEAR II is surrounded with coniferous forest and
has limited anthropogenic emission sources nearby. Diverse measurements of meteorology, aerosol
and gas phase properties are continuously conducted at the station. Details about the meteorological
conditions and temporal variations of trace gases during IBAIRN campaign are presented by Zha et
al. (2018) and Liebmann et al. (2018).
2.1.2 Instrument and data
Data were collected with a nitrate ($NO_3^-$)-based chemical ionization atmospheric pressure interface
time-of-flight mass spectrometer (CI-APi-TOF, Jokinen et al., 2012) with about 4000 Th $Th^{-1}$ mass
resolving power, at ground level in September, 2016. In our study, the mass spectra were averaged to
1 h time resolution from September $6^{th}$ to $22^{nd}$ for further analysis. We use the thomson (Th) as the



unit for mass/charge, with 1 Th = 1 Da/e, where $e$ is the elementary charge. As all the data discussed
in this work are based on negative ion mass spectrometry, we will use the absolute value of the
mass/charge, although the charge of each ion will be negative. The masses discussed in this work
includes the contribution from the nitrate ion, 62, unless specifically mentioned. Furthermore,  as the
technique is based on soft ionization with $NO_3^-$ ions, any multiple charging effects are unlikely, and
therefore the reported mass/charge values in thomson can be considered equivalent to the mass of the
ion in Da.
The forest site of Hyytiälä is dominated by monoterpene emissions (Hakola et al., 2006). The main
feature of previous CI-APi-TOF measurements in Hyytiälä (Ehn et al., 2014;Yan et al., 2016) has
been a bimodal distributions of HOM, termed monomers and dimers, as they are formed of either one
or two $RO_2$ radicals, respectively. For the analysis in this study, we chose three mass/charge ($m/z$)
ranges of 50 Th each (Figure 1), corresponding to regions between which we expect differences in
formation or loss mechanisms. In addition to regions with HOM monomers and HOM dimers, one
range was chosen at lower masses, in a region presumably mainly consisting of molecules that are
less likely to condense onto aerosol particles (Peräkylä et al., 2019).
2.2 Positive matrix factorization (PMF)
After the model of PMF was developed (Paatero and Tapper, 1994), numerous applications have been
conducted with different types of environmental data (Song et al., 2007;Ulbrich et al., 2009;Yan et
al., 2016;Zhang et al., 2017). By reducing dimensionality of the measured dataset, PMF model greatly
simplifies the data analysis process with no requirement for prior knowledge of sources or pathways
as essential input. The main factors can be further interpreted with their unique/dominant markers
(elements or masses).
The basic assumption for PMF modelling is mass balance, which assumes that ambient concentration
of a chemical component is the sum of contributions from several sources or processes, as shown in
equation (1).
$$\mathbf{X} = \mathbf{TS} \times \mathbf{MS} + \mathbf{R} \tag{1}$$
In equation (1), $\mathbf{X}$ stands for the time series of measured concentration of different variables ($m/z$ in
our case), $\mathbf{TS}$ represents the temporal variation of factor contributions, $\mathbf{MS}$ stands for factor profiles
(mass spectral profiles), and $\mathbf{R}$ is the residual as the difference of the modelled and the observed data.
The matrices $\mathbf{TS}$ and $\mathbf{MS}$ are iteratively calculated by a least-squares algorithm utilizing uncertainty
estimates, to pursue minimized $Q$ value as shown in equation (2), where $S_{ij}$ is the estimated
uncertainty, an essential input in PMF model.
$$Q = \sum \sum (\frac{R_{ij}}{S_{ij}})^2 \tag{2}$$



PMF model was conducted by multi-linear engine (ME-2) (Paatero, 1999) interfaced with Source
Finder (SoFi, v6.3) (Canonaco et al., 2013). Signal-to-noise ratio (SNR) was calculated as $SNR_{ij} =$
abs ($Xij$) / abs ($Sij$). When the Signal-to-noise ratio (SNR) is below 1, the signal of $X_{ij}$ will be down-
weighted by replacing the corresponding uncertainty $S_{ij}$ by $S_{ij}/SNR_{ij}$ (Visser et al., 2015). Future
studies should pay attention to the potential risk when utilizing this method since down-weighting
low signals element-wise will create a positive bias to the data. Robust mode was operated in the
PMF modelling, where outliers ($\left|\frac{R_{ij}}{S_{ij}}\right| > 4$) were significantly down-weighted (Paatero, 1997).
2.3 binPMF
As a newly developed application of PMF for mass spectral data, binPMF has no requirement for
chemical composition information, while still taking advantage of the HR mass spectra, saving effort
and time (Zhang et al., 2019). To explore the benefits of analyzing separated mass ranges, we applied
binPMF to the three separated ranges. The three ranges were also later combined for binPMF analysis
as comparison with the previous results. The PMF model requires both data matrix and error matrix
as input, and details of the preparation of data and error matrices are described below.
2.3.1 Data matrix
Different from normal UMR or HR peak fitting, in binPMF, the mass spectra are divided into small
bins after baseline subtraction and mass axis calibration. Linear interpolation was first conducted to
the mass spectra with a mass interval of 0.001 Th. Then the interpolated data was averaged into bins
of 0.02 Th width. We selected three ranges for further analysis based on earlier studies (Ehn et al.,
2014;Yan et al., 2016;Bianchi et al., 2019;Peräkylä et al., 2019).
- Range 1, *m/z* 250 – 300 Th, 51 unit masses × 25 bins per unit mass = 1275 bins/variables,
consisting mainly of molecules with five to nine carbon atoms and four to nine oxygen atoms
in our dataset.
- Range 2, *m/z* 300 – 350 Th, 51×25 = 1275 bins, mainly corresponding to HOM monomer
products, featured with nine to ten C- and seven to ten O-atoms.
- Range 3, *m/z* 510 – 560 Th, 51×30 = 1530 bins, mainly corresponding to HOM dimer products,
with carbon numbers of sixteen to twenty and eleven to fifteen O-atoms.
For a nominal mass *N*, the signal region included in further analyses was between *N*-0.2 Th and *N*+0.3
Th for Range 1 and 2, and between *N*-0.2 Th and *N*+0.4 Th for Range 3. The data were averaged into
1-h time resolution and in total we had 384 time points in the data matrix.
2.3.2 Error matrix



The error matrix represents the estimated uncertainty for each element of the data matrix and is crucial
for iterative calculation of the $Q$ minimum. Equation (3) is used for error estimation (Polissar et al.,

1998),

$$S_{ij} = \sigma_{ij} + \sigma_{\text{noise}} \tag{3}$$

where $S_{ij}$ represents the uncertainty of $m/z$ $j$ at time $i$, $\sigma_{ij}$ stands for counting statistics uncertainty
and is estimated as follows,

$$\sigma_{ij} = a \times \frac{\sqrt{I_{ij}}}{\sqrt{t_s}} \tag{4}$$

where $I$ is the signal intensity term, in unit of counts per second (cps), $t_s$ stands for length of averaging
in seconds, while $a$ is an empirical coefficient to compensate for unaccounted uncertainties (Allan et
al., 2003;Yan et al., 2016) and is 1.28 in our study as previously estimated from laboratory
experiments (Yan et al., 2016). The $\sigma_{noise}$ term was estimated as the median of the standard
deviations from signals in the bins in the region between nominal masses, where no physically
meaningful signals are expected.
**3 Results**
3.1 General overview of the dataset/spectrum
During the campaign, in autumn, 2016, the weather was overall sunny and humid with average
temperature of 10.8 °C and relative humidity (RH) of 87% (Zha et al., 2019). The average
concentration of $NO_x$ and $O_3$ was 0.4 ppbv and 21 ppbv, respectively. The average total HOM
concentration was ~ $10^8$ molecules $cm^{-3}$.

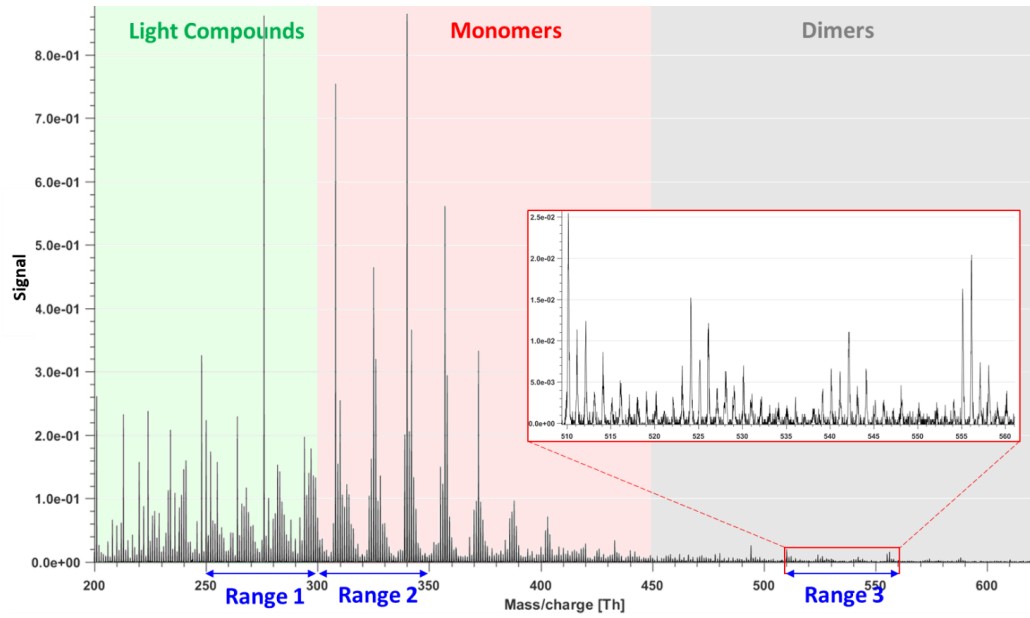




Figure 1. Example of mass spectrum with 1-h time resolution measured from a boreal forest
environment during the IBAIRN campaign (at 18:00, Finnish local time, UTC+2). The mass
spectrum was divided into three parts and three sub-ranges were chosen from different parts for
further analysis in our study.  The nitrate ion (62 Th) is included in the mass.
Figure 1 shows the 1 h averaged mass spectrum taken at 18:00 on September 12, as an example of
the analyzed dataset. In addition to exploring the benefits of this type of sub-range analysis in relation
to different formation or loss pathways, separating into sub-ranges may also aid factor identification
for low-signal regions. As shown in Figure 1, there is a difference of 1-2 orders of magnitude in the
signal intensity between Range 3 and Ranges 1-2. If all Ranges are run together, we would expect
that the higher signals from Ranges 1 and 2 will drive the factorization. While if run separately,
separating formation pathways of dimers in Range 3 will likely be easier. As dimers have been shown
to be crucial for the formation of new aerosol particles from monoterpene oxidation (Kirkby et al.,
2016;Troestl et al., 2016;Lehtipalo et al., 2018), this information may even be the most critical in
some cases, despite the low contribution of these peaks to the total measured signal.
binPMF was separately applied to Range 1, 2, 3, and a 'Range combined' which comprised all the
three sub-ranges. All the PMF runs for the four ranges were conducted from two to ten factors and
repeated three times for each factor number, to assure the consistency of the results. Factorization
results and evolution with increasing factor number are briefly described in the following sections,
separately for each Range (sections 3.2 – 3.5). More detailed discussion and comparison between the
results are presented in Section 4.
3.2 binPMF on Range 1 (250 – 300 Th)
As has become routine (Zhang et al., 2011;Craven et al., 2012), we first examined the mathematical
parameters of our solutions. From two to ten factors, $Q/Q_{exp}$ decreased from 2.8 to 0.7 (Fig S1 in
supplementary information), and after three factors, the decreasing trend was gradually slowing down
and approaching one, which is the ideal value for $Q/Q_{exp}$ as a diagnostic parameter. The unexplained
variation showed a decline from 18% to 8% from two to ten factors.
In the two-factor results, two daytime factors were separated, with peak time both at 14:00 - 15:00.
One factor was characterized by large signals at 250 Th, 255 Th, 264 Th, 281 Th, 283 Th, 295 Th,
Th. The other factor was characterized by large signals at 294 Th, 250 Th, 252 Th, 264 Th, 266
Th, 268 Th, and 297 Th. In Hyytiälä, as reported in previous studies, odd masses observed by the
nitrate CI-APi-TOF are generally linked to monoterpene-derived organonitrates during the day (Ehn
et al., 2014;Yan et al., 2016). When the factor number increased to three, the two earlier daytime
factors remained similar with the previous result, while a new factor appeared with a distinct sawtooth
shape in the diurnal cycle. The main marker in the spectral profile was 276 Th, with a clear negative



mass defect. When one more factor was added, the previous three factors remained similar as in the
three-factor solution, and a new morning factor was resolved, with 264 Th and 297 Th dominant in
the mass spectral profile, and a diurnal peak at 11:00.
As the factor number was increased, more daytime factors were separated, with similar spectral
profiles to existing daytime factors and various peak times. No nighttime factors were found in the
analysis even when the factor number reached ten. We chose the four-factor result for further
discussion, and Figure 2 shows the result of Range 1, with spectral profile, time series, diurnal cycle
and factor contribution. As shown in Figure 2d, Factors 1-3 are all daytime factors, while Factor 4
has a sawtooth shape, which is caused by contamination, mainly by perfluorinated acids, of the inlet's
automated zeroing every three hours during the measurements (Zhang et al., 2019). The zeroing
periods have been removed from the dataset before binPMF analysis, but the contamination factor
was still resolved. This factor is discussed in more detail in sections 4.1.1 and 4.1.4.

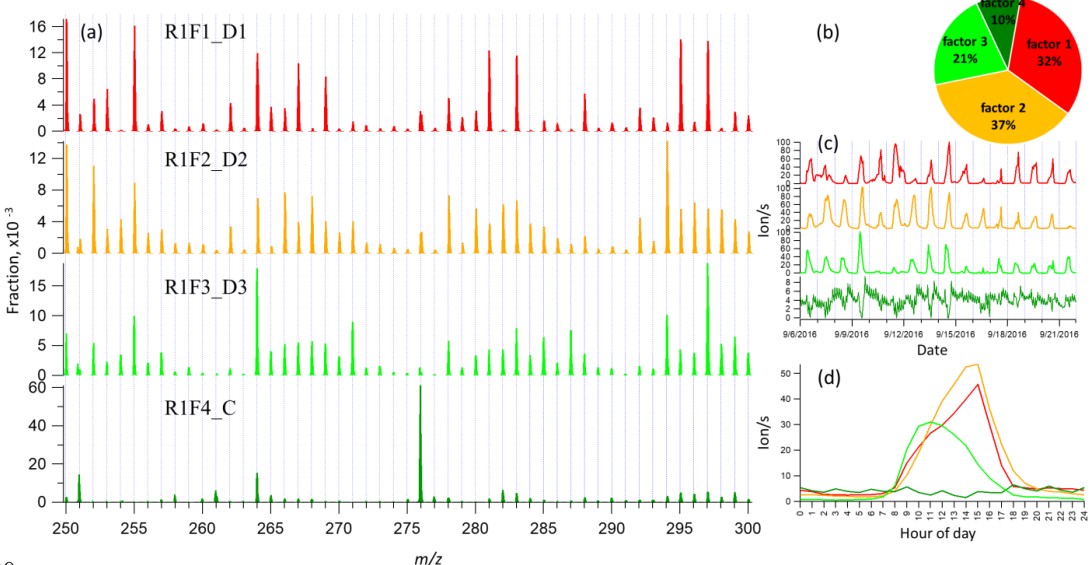


Figure 2 Four-factor result for Range 1, for (a) factor spectral profiles, (b) factor contribution, (c)
time series and (d) diurnal trend. Details on the factors' naming schemes are shown in Table 1.
3.3 binPMF on Range 2 (300-350 Th)
This range covers the monoterpene HOM monomer range, and binPMF results have already been
discussed by Zhang et al. (2019) as a first example of the application of binPMF on ambient data.
Our input data here is slightly different. In the previous study, the 10 min automatic zeroing every
three hours was not removed before averaging to 1 hour time resolution while here, we have removed
this data. Overall, the results are similar as in our earlier study, and therefore the result are just briefly



summarized below for further comparison and discussion in Section 4. Similar to Range 1, both the
$Q/Q_{exp}$ (2.2 to 0.6) and unexplained variation (16% to 8%) declined with the increased factor number
from two to ten.
When the factor number was two, one daytime and one nighttime factor were separated, with diurnal
peak times at 14:00 and 17:00, respectively. The nighttime factor was characterized by masses at 340
Th, 308 Th and 325 Th (monoterpene ozonolysis HOM monomers (Ehn et al., 2014)) and remained
stable throughout the factor evolution from two to ten factors. With the addition of more factors, no
more nighttime factors got separated while the daytime factor was further separated and more daytime
factors appeared, peaking at various times in the morning (10:00 am), at noon or in the early afternoon
(around 14:00 pm and 15:00 pm). High contribution of 339 Th can be found in all the daytime factor
profiles. As the factor number reached six, a contamination factor appeared, characterized by large
signals at 339 Th and 324 Th, showing negative mass defects (Figure S2 in the Supplement). The
factor profile is nearly identical to the contamination factor determined in Zhang et al. (2019), where
the zeroing periods were not removed, causing larger signals for the contaminants. In our dataset,
where the zeroing periods were removed, no sawtooth pattern was discernible in the diurnal trend,
yet it could still be separated even though it only contributed 3% to Range 2. More about the
contamination factors from different ranges will be discussed in Section 4.1.4. Since the aim of this
study is mainly to explore the benefits of analyzing different ranges of the mass spectrum, we chose
to show the four-factor result below, to simplify the later discussion and comparison. Figure 3 shows
four-factor result of Range 2, with spectral profile, time series, diurnal cycle and factor contribution.

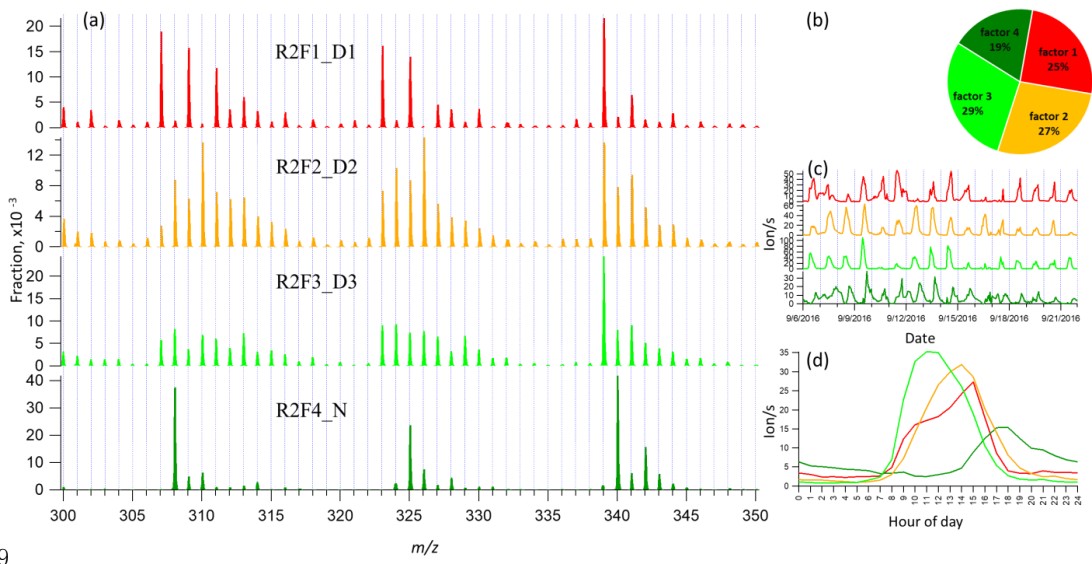




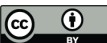

Figure 3 Four-factor result for Range 2, for (a) factor spectral profiles, (b) factor contribution, (c)
time series and (d) diurnal trend. Details on the factors' naming schemes are shown in Table 1.
3.4 binPMF on Range 3 (510-560 Th)
Range 3 represents mainly the monoterpene HOM dimers (Ehn et al., 2014). Similar to Range 1 and
2, both the $Q/Q_{exp}$ (1.5 to 0.6) and unexplained variation (18% to 15%) showed decreasing trend with
the increased factor number (2-10). As can be seen from Figure 1, data in Range 3 had much lower
signals, compared to that of the Range 1 and 2, explaining the higher unexplained variation for Range

337 3.

In the two-factor result for Range 3, one daytime and one nighttime factor appeared, with diurnal
peak times at noon and 18:00, respectively. The nighttime factor was characterized by ions at 510 Th,
Th, 526 Th, 542 Th, and 555 Th, 556 Th, while the daytime factor showed no dominant marker
masses, yet with relatively high signals at 516 Th, 518 Th and 520 Th. As the number of factors
increased to three, one factor with almost flat diurnal trend was separated, with dominant masses of
510 Th, 529 Th, 558 Th. Most peaks in this factor had negative mass defects, and this factor was
again linked to a contamination factor. The four-factor result resolved another nighttime factor with
a dominant peak at 555 Th, and effectively zero contribution during daytime. As the factor number
was further increased, the new factors seemed like splits from previous factors with similar spectral
profiles. We therefore chose four-factor result also for Range 3 (results shown in Fig. 4) for further
discussion.

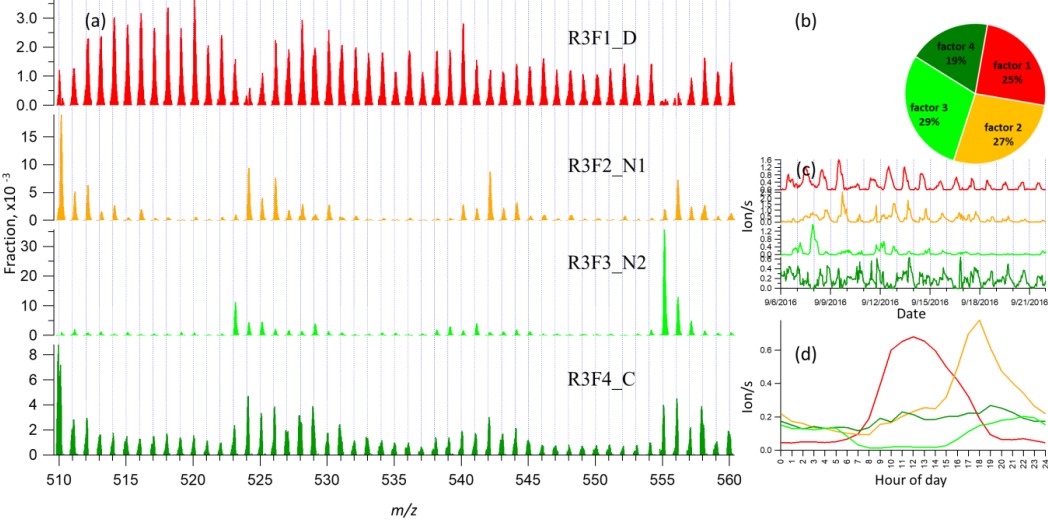


Figure 4 Four-factor result for Range 3, for (a) factor spectral profiles, (b) factor contribution, (c)
time series and (d) diurnal trend. Details on the factors' naming schemes are shown in Table 1.

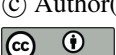



3.5 binPMF on Range Combined (250-350 Th & 510-560 Th)
As comparison to the previous three ranges, we conducted the binPMF analysis on Range Combined,
which is the combination of the three ranges. The results of this range are fairly similar to those of
Ranges 1 and 2, as could be expected since the signal intensities in these ranges were much higher
than in Range 3. As the number of factors increased (2-10), both the $Q/Q_{exp}$ (1.3 to 0.6) and
unexplained variation (16% to 8%) showed a decreasing trend.
In the two-factor result, one daytime factor and one nighttime factor were separated. In the nighttime
factor, most masses were found at even masses, and the fraction of masses in Range 3 was much
higher than that in daytime factor. In contrast, in the daytime factor, most masses were observed at
odd masses and the fraction of signal in Range 3 was much lower. During the day, photochemical
reactions increase the concentration of NO, which serves as peroxy radical (RO$_2$) terminator and often
outcompetes RO$_2$ cross reactions in which dimers can be formed (Ehn et al., 2014). Thus, the
production of dimers is suppressed during the day, yielding instead a larger fraction of organic nitrates,
as has been shown also previously (Yan et al., 2016).
With the increase of the number of factors, more daytime factors were resolved with different peak
times. When the factor number reached seven, a clear sawtooth-shape diurnal cycle occurred, i.e. the
contamination factor, caused by the zeroing. As more factors were added, no further nighttime factors
were separated, and only more daytime factors appeared. To simplify the discussion and inter-range
comparison, we also here chose the four-factor result for further analysis, as it already provided
enough information for our main goal in this study. Figure 5 shows the four-factor result of Range
Combined, with spectral profile, time series, diurnal cycle and factor contribution. The signals in
range of 510-560 Th were enlarged 100-fold to be visible.

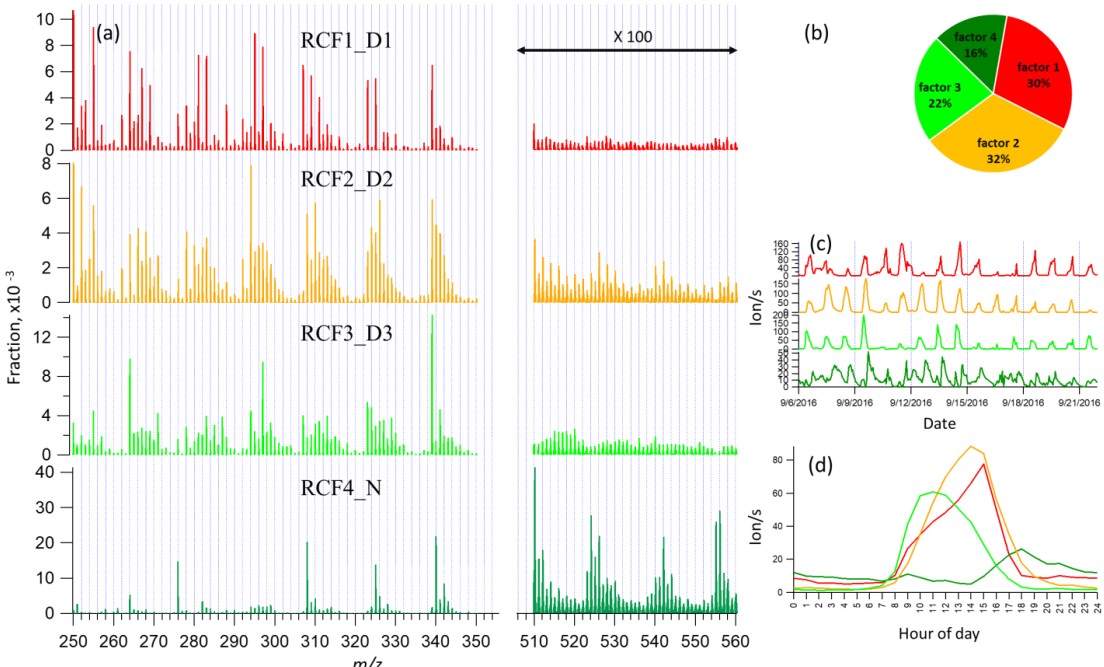


Figure 5 Four-factor result for Range Combined, for (a) factor spectral profiles, (b) factor

contribution, (c) time series and (d) diurnal trend. Details on the factors' naming schemes are shown

in Table 1.

## 4 Discussion

In Section 3, results by binPMF analysis were shown for Ranges 1, 2, 3 and Combined. In this section,

we discuss and compare the results from the different ranges. To simplify the inter-range comparison,

we chose four-factor results for all the four ranges, with the abbreviations shown in Table 1. From

Range 1, three daytime factors and a contaminations factor were separated. In Range 2, three daytime

factors and one nighttime factor (abbreviated as R2F4_N) were resolved. The R2F4_N factor was

characterized by signals at 308 Th ($C_{10}H_{14}O_7 \cdot NO_3^-$), 325 Th ($C_{10}H_{15}O_8 \cdot NO_3^-$), and 340 Th

($C_{10}H_{14}O_9 \cdot NO_3^-$), and can be confirmed as monoterpene ozonolysis products (Ehn et al., 2014;Yan et

al., 2016). With the increase of factor number to six, the contamination factor got separated also in

this mass range. In Range 3, one daytime factor, two nighttime factors and a contamination factor

were separated. The first nighttime factor (R3F2_N1) had large peaks at 510 Th ($C_{20}H_{32}O_{11} \cdot NO_3^-$)

and 556 Th ($C_{20}H_{30}O_{14} \cdot NO_3^-$), dimer products that have been identified during chamber studies of

monoterpene ozonolysis (Ehn et al., 2014). The molecule observed at 510 Th has 32 H-atoms,

suggesting that one of the $RO_2$ involved would have been initiated by OH, which is formed during

the ozonolysis of alkenes such as monoterpenes at nighttime (Atkinson et al., 1992;Paulson and





Orlando, 1996). The other nighttime factor (R3F3_N2) was dominated by ions at 523 Th
($C_{20}H_{31}O_8NO_3 \cdot NO_3^-$) and 555 Th ($C_{20}H_{31}O_{10}NO_3 \cdot NO_3^-$), representing nighttime monoterpene
oxidation involving $NO_3$. As these dimers contain only one N-atom, and 31 H-atoms, we can assume
that they are formed from reactions between an $RO_2$ formed from $NO_3$ oxidation and another $RO_2$
formed by ozone oxidation. These results match well with the profiles in a previous study by Yan et
al. (2016). The results of Range Combined are very similar to Range 2, with one nighttime factor and
three daytime factors. The contamination factor was separated with increase of factor number to seven.

Table 1. Summary of PMF results for the different mass ranges

| Range | Factor number | Factor name[a] | Dominant peaks | Peak time |
|---|---|---|---|---|
| **1 (250 - 300 Th)** | 1 | R1F1_D1 | 250, 255, 295, 297 | 15:00 |
| | 2 | R1F2_D2 | 250, 252, 294 | 15:00 |
| | 3 | R1F3_D3 | 264, 297 | 11:00 |
| | 4 | R1F4_C | 276 | -[b] |
| **2 (300 - 350 Th)** | 1 | R2F1_D1 | 307, 309, 323, 325, 339, | 15:00 |
| | 2 | R2F2_D2 | 310, 326, 339, | 14:00 |
| | 3 | R2F3_D3 | 339 | 11:00 |
| | 4 | R2F4_N | 308, 325, 340 | 18:00 |
| **3 (510 – 560 Th)** | 1 | R3F1_D | 516, 518, 520, 528, 540 | 12:00 |
| | 2 | R3F2_N1 | 510, 524, 542, 556 | 18:00 |
| | 3 | R3F3_N2 | 523, 555 | 22:00 |
| | 4 | R3F4_C | 510, 558 | -[b] |
| **Combined (1, 2, 3)** | 1 | RCF1_D1 | 250, 255, 295, 339 | 15:00 |
| | 2 | RCF2_D2 | 250, 252, 294, 339 | 14:00 |
| | 3 | RCF3_D3 | 264, 297, 339 | 11:00 |
| | 4 | RCF4_N | 308, 340, 510, 524, 555, 556 | 18:00 |

[a] Factor name is defined with range name, factor number and name. For example, RxFy represents Factor y in Range x.
RC stands for Range Combined. For the factor name, D is short for daytime, N for Nighttime, C for contamination.
[b] The contamination factor in Range 1 shows sawtooth pattern; while in Range 3 shows no diurnal pattern.

4.1 Comparison of different ranges
4.1.1 Time series correlation
In Figure 6, the upper panels show the time series correlations among the first three ranges. As
expected based on the results above, generally the daytime factors, and the two nighttime
monoterpene ozonolysis factors (R2F4_N and R3F2_N1) correlated well, respectively. However, the
contamination factors did now show strong correlation between different ranges, even though
undoubtedly from the same source. More about the contamination factors will be discussed in Section
4.1.4. The lower panels in Figure 6 displays the correlations between the first three ranges and the
Range Combined, and clearly demonstrates that the results of Range Combined is mainly controlled
by high signals from Range 1 and 2. More detailed aspects of the comparison between factors in



different ranges is given in the following sections. The good agreements between factors from
different subranges also help to verify the robustness of the solutions.

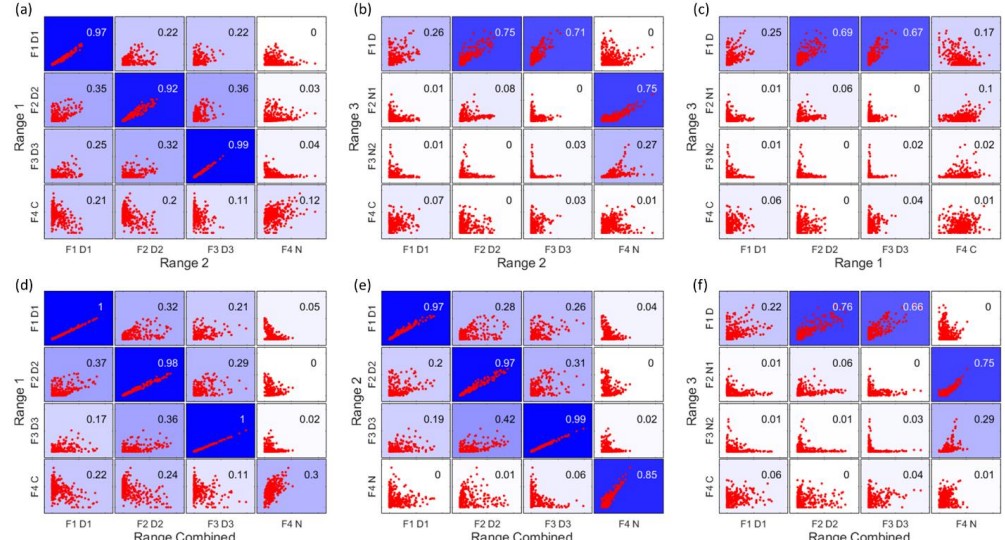

Figure 6 Time series correlations among Range 1, 2, 3 (upper panels a-c), and between the first three
ranges and the Range Combined (lower panels d-f). The abbreviations for different factors are the
same in Table 1, with F for factor, D for daytime, N for nighttime and C for contamination, e.g. F1D1
for Factor 1 daytime 1. The coefficient of determination, $R^2$, is marked in each subplot by a number
shown in the right upper corners and by the blue colors, with stronger blue indicating higher $R^2$.
4.1.2 Daytime factor comparison
As mentioned above, with increasing number of factors, usually more daytime factors will be resolved,
reflecting the complicated daytime photochemistry. The three daytime factors between Range 1 and
2 agreed with each other quite well (Figure 6a). However, R1F1_D1 and R2F1_D1 did not show
strong correlation with the only daytime factor in Range 3 (R3F1_D), while the other two daytime
factors in both Range 1 and 2, i.e. R1F2_D2, R1F3_D3, R2F2_D2, R2F3_D3, correlated well with
R3F1_D from Range 3.
The 1st daytime factors from Range 1 and 2, R1F1_D1 and R2F1_D1, were mainly characterized by
odd masses 255 Th, 281 Th, 283 Th, 295 Th, 297 Th, 307 Th, 309 Th, 311 Th, 323 Th, 325 Th, 339
Th. The factors are dominated by organonitrates. Organic nitrate formation during daytime is
generally associated with the termination of $RO_2$ radicals by NO. This termination step is mutually
exclusive with the termination of $RO_2$ with other $RO_2$, leading to dimer formation. If the NO
concentration is the limiting factor for the formation of these factors, the low correlations between
the NO-terminated monomer factors, and the dimer factors, is to be expected. In contrast, if the other



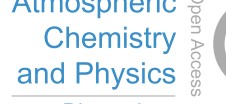
daytime factors mainly depend on oxidant and monoterpene concentrations, some correlation
between those, and the daytime dimer factor, is to be expected, as shown in Figure 6b, c.
All the spectral profiles resolved from Range Combined binPMF analysis inevitably contained mass
contribution from 510 – 560 Th, even the daytime factor from Range Combined (RCF1_D1) which
did not show clear correlation with R3F1_D from Range 3 (Figure 6e).
The 2$^{nd}$ and 3$^{rd}$ daytime factors in Range 1 and 2, R1F2_D2, R1F3_D3, R2F2_D2, R2F3_D3, had
high correlations with R3F1_D in Range 3. Daytime factors in Range Combined (RCF2_D2 and
RCF3_D3) also showed good correlation with R3F1_D in Range 3. However, if we compare R3F1_D
and the mass range of 510 – 560 Th of the daytime factors in Range Combined, just with a quick look,
we can readily see the difference. The daytime factor separated in Range 3 (R3F1_D) basically has
no obvious markers in the profile, and as mentioned above, up to ten factors, there would only be
more factors fragmented from the previous factor, with similar spectral profiles, but showed different
profile pattern with 510 – 560 Th in RCF2_D2 in Range Combined. The factorization of Range
Combined was mainly controlled by Range 1 and 2 due to high signals, and the signals in Range 3
are forced to be distributed according to the time series determined by Ranges 1 and 2. Ultimately,
this will lead to failure in factor separation for this low-signal range.
4.1.3 Nighttime factor comparison
Since high-mass dimers are more likely to form at night due to photochemical production of NO in
daytime, which inhibits $RO_2$ + $RO_2$ reactions, Range 3 had the highest fraction of nighttime signals
of all the sub-ranges. While Range 3 produced two nighttime factors, Ranges 2 and Combined showed
one, and Range 1 had no nighttime factor. The difference between the two results also indicates the
advantage of analyzing monomers and dimers separately.
The two nighttime factors in Range 3 can be clearly identified as arising from ozonolysis (R3F2_N1)
and a mix of ozonolysis and $NO_3$ oxidation (R3F2_N2) based on the mass spectral profiles, as
described above. The organonitrate at 555 Th, $C_{20}H_{31}O_{10}NO_3 \cdot NO_3^-$, is a typical marker for $NO_3$
radical initiated monoterpene chemistry (Yan et al., 2016). However, several interesting features
become evident when comparing to the results of Range 2 and Combined. Firstly, only one nighttime
factor (R2F4_N, RCF4_N) was separated in each of these ranges, and that shows clear resemblance
with ozonolysis of monoterpenes as measured in numerous studies, e.g. (Ehn et al., 2012;Ehn et al.,
2014). Secondly, the high correlation found in Figure 6b between the ozonolysis factors (i.e.,
R2F4_N, R3F2_N1, RCF4_N), further supports the assignment. However, this factor is the only
nighttime factor in the monomer range, suggesting that $NO_3$ radical chemistry of monoterpenes in
Hyytiälä does not form substantial amounts of HOM monomers. The only way for the CI-APi-TOF
to detect products of monoterpene-$NO_3$ radical chemistry may thus be through the dimers, where one





highly oxygenated $RO_2$ radical from ozonolysis reacts with a less oxygenated $RO_2$ radical from $NO_3$
oxidation.
In the results by Yan et al. (2016) the combined UMR-PMF of monomers and dimers did yield a
considerable amount of compounds in the monomer range also for the $NO_3$ radical chemistry factor.
There may be several reasons for this discrepancy. One major cause for differences between the spring
dataset of Yan et al. (2016) and the autumn dataset presented here, is that nighttime concentrations
of HOM was greatly reduced during our autumn campaign. The cause may have been fairly frequent
fog formation during nights, and also the concentration of e.g. ozone decreased nearly to zero during
several nights (Zha et al., 2018). It is also possible that the $NO_3$ radical-related factor by Yan et al.
(2016) is probably a mixture of $NO_3$ and $O_3$ radical chemistry, while the monomer may thus be
attributed to the $O_3$ part. Alternatively, the different conditions during the two measurement periods,
as well as seasonal difference in monoterpene mixtures (Hakola et al., 2012), caused variations in the
oxidation pathways.
4.1.4 Contamination factor
During the campaign, an automated instrument zeroing every three hours was conducted, by
switching a valve to pass the air through a HEPA filter. Each zeroing process lasted for 10 min. While
the zeroing successfully removed the low-volatile HOM and $H_2SO_4$, the zeroing process introduced
contaminants into the inlet lines. The contaminants were primarily different types of perfluorinated
organic acids, often off-gassing from e.g. Teflon tubing. For IVOC contaminants, these would be
flushed through the inlet, while (E)LVOC would condense onto the inlet walls and not come off.
However, SVOC contaminants may stick to the inlet tubing and slowly evaporate back into the
sampled air. We removed all the 10-min zeroing periods, and averaged the data to 1-h time resolution,
but contaminants were still identified in all ranges by binPMF.
Contamination contributed 10%, 3%, 19% and 4% to Range 1, 2, 3, and Combined, respectively, in
the binPMF solutions where the contamination factor was first separated. This also explains why the
contamination factor was separated much earlier in Ranges 1 and 3 than in Range 2. However, despite
contributing slightly more to Range Combined than to Range 2, the contamination factor was
separated when the factor number was increased by one in Range Combined. Here, the difference in
volatility of the contaminants in the different sub-ranges may play a role, such that the contaminants
in different sub-ranges behave differently. Thus, the behavior of the contamination factor across the
combined range is not consistent. Therefore, we examined the zeroing effect with finer time
resolution, i.e. 1 min, with three of the largest fluorinated compounds in each range of our mass
spectrum, $(CF_2)_3CO_2HF \cdot NO_3^-$ (275.9748 Th), $(CF_2)_5C_2O_4H^-$ (338.9721 Th), and $(CF_2)_6CO_2HF \cdot NO_3^-$
(425.9653 Th). Since the overall signal levels were very low for these compounds, the time series



became very noisy with such high time resolution. This made it impossible to perform HR fitting for
the data, and instead we summed up the signal from the mass ranges where we expected unperturbed
signal from these ions.
The time series with sawtooth pattern of the three fluorinated compounds is shown in Figure S3 in
Supplement. From the time series, we selected a period of around three days of the 3-h cycles (25 in
total), and in Figure 7 the cycles were aligned and superimposed on top of one another, normalized
by the maximum during the zeroing. The normalized signals of the three compounds are shown in
light colors, and the mean values shown in bold sold lines. This data includes also the zeroing periods
to highlight the effect, but these periods were removed from the data used for our PMF analyses.

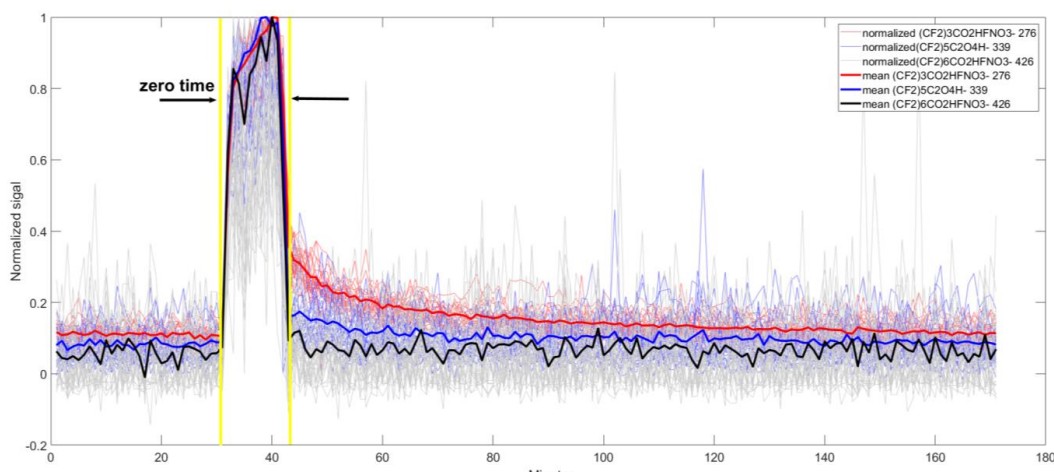

Figure 7 Normalized signals for three fluorinated compounds during a 3-h cycle (180 minutes), with
$(CF_2)_3CO_2HF \cdot NO_3^-$ (275.9748 Th) in red, $(CF_2)_5C_2O_4H^-$ (338.9721 Th) in blue, and
$(CF_2)_6CO_2HF \cdot NO_3^-$ (425.9653 Th) in black. We selected 25 cycles and normalized all the cycles by
their individual maximum. The yellow window shows the zeroing time, for around 10 minutes, which
has been removed from the data analysis. Light colors display the individual cycles, and the bold
solid colors stand for the average for each compounds.
The signals of the three fluorinated compounds increased by 10 to 20 times during the zeros, due to
off-gassing either in the filter or in the tubing in the zeroing setup. Immediately after the zeroing was
stopped, signals of all three compounds dropped by about 60-90%, followed by a gradual decay. The
decay period coincided with our ambient sampling, and therefore these signals are part of our dataset.
It is evident that the three fluorinated compounds were from the same source (zeroing process), but
due to their different volatilities, they were lost at different rates. This, in turn, means that the spectral
signature of this source will change as a function of time, at odds with one of the basic assumptions





of PMF. Panels a and b in Figure S4 displays the temporal correlation with and without zeroing period
with 1 min time resolution. The correlation coefficients dropped greatly when the zero period was
removed, from 0.9 to 0.3 for $R^2$ between 276 Th and 339 Th, and 0.8 to 0.1 between 276 Th and 426
Th. Similar effect is also found with the 1 h averaged data (Fig. S4c, d).
This detailed analysis of fluorinated contamination in our system was here merely used as an example
to show that volatility can impact source profiles over time. In this case, the contamination factor was
still identified both from the separate sub-ranges and from the combined data set using binPMF.
However, the contamination profile in the combined range is now averaged, compared to that from
separate ranges: the fractional contributions of contamination compounds to this profile, vary during
the process of each zeroing due to different volatility properties. In Figure S5, contamination factor
profiles from Range 3 and Range Combined were compared. It can be clearly seen that the profile of
Range Combined is more noisy than that of Range 3, probably due to the varied fractional
contributions of contamination compounds to the profile. In ambient data, products from different
sources can have undergone atmospheric processing, altering the product distribution. Our aim with
this analysis was to highlight the importance of differences in the sink terms due to different
volatilities of the products. This may be an important issue for gas phase mass spectrometry analysis,
potentially underestimated by many PMF users, as it is likely only a minor issue for aerosol data, for
which PMF has been applied much more routinely. If failing to achieve physically meaningful factors
using PMF on gas phase mass spectra, our recommendation is to try applying PMF to sub-ranges of
the spectrum, where IVOC, SVOC and (E)LVOC could be analyzed separately.
4.2 Atmospheric insights
While the previous section discussed several findings with atmospheric implications, we highlight
two results below, which are particularly intriguing. We also include the correlation matrix of all
PMF and factors and selected meteorological parameters in Table 2.

Table 2 Correlation between factors and meteorological parameters and gases

| | R1F1_D1 | R1F1_D2 | R1F1_D3 | R1F1_C | R2F1_D1 | R2F2_D2 | R2F3_D3 | R2F4_N | R3F1_D | R3F2_N1 | R3F3_N2 | R3F4_C | RCF1_D1 | RCF2_D2 | RCF3_D3 | RCF4_N |
|---|---|---|---|---|---|---|---|---|---|---|---|---|---|---|---|---|
| O₃ | 0.51 | 0.59 | 0.35 | -0.18 | 0.47 | 0.57 | 0.36 | 0.43 | 0.55 | 0.33 | 0.27 | 0.22 | 0.49 | 0.57 | 0.33 | 0.34 |
| NO | 0.13 | -0.01 | 0.24 | -0.03 | 0.18 | -0.02 | 0.24 | -0.22 | 0.13 | -0.19 | -0.17 | 0.03 | 0.13 | 0.00 | 0.26 | -0.18 |
| NOx | -0.05 | -0.22 | -0.10 | 0.09 | -0.01 | -0.23 | -0.11 | -0.13 | -0.16 | -0.21 | -0.04 | 0.04 | -0.04 | -0.22 | -0.09 | -0.11 |
| RH | -0.46 | -0.80 | -0.63 | 0.30 | -0.43 | -0.82 | -0.64 | -0.27 | -0.78 | -0.39 | -0.07 | -0.07 | -0.43 | -0.82 | -0.60 | -0.21 |
| T | 0.66 | 0.72 | 0.40 | -0.24 | 0.65 | 0.66 | 0.41 | 0.39 | 0.65 | 0.30 | 0.14 | 0.19 | 0.66 | 0.68 | 0.38 | 0.24 |
| UVB | 0.52 | 0.63 | 0.82 | -0.40 | 0.52 | 0.68 | 0.84 | -0.30 | 0.79 | -0.08 | -0.27 | 0.08 | 0.49 | 0.68 | 0.83 | -0.29 |







### 4.2.1 Daytime dimer formation

Dimers are primarily produced during nighttime, due to NO suppressing $RO_2 + RO_2$ reactions in daytime (Ehn et al., 2014;Yan et al., 2016). However, in this study, we found one clear daytime factor in Range 3 (R3F1_D, peak at local time 12:00, UTC+2) by sub-range analysis. With high loadings from even masses including 516, 518, 520, 528, 540 Th, this only daytime factor in dimer range correlated very well with two daytime factors in Ranges 1 and 2 (R1F2_D2, R1F3_D3, R2F2_D2, R2F3_D3) (Figure 6b and c). Strong correlation between R3F1_D with solar radiation was found, with R = 0.79 (Table 2). This may indicate involvement of OH oxidation in producing this factor.

As previous studies have shown, dimers greatly facilitate new particle formation (NPF) (Kirkby et al., 2016;Troestl et al., 2016;Lehtipalo et al., 2018), and this daytime dimer factor may represent a source of dimers that would impact the initial stages of NPF in Hyytiälä. Mohr et al. (2017) reported a clear diel pattern of dimers (sum of about 60 dimeric compounds of $C_{16-20}H_{13-33}O_{6-9}$) during NPF events in 2013 in Hyytiälä, with minimum at night and maximum after noon, and estimated these dimers can contribute ~5% of the mass of sub-60 nm particles. The link between the dimers presented in that paper and those reported here will require further studies, as will the proper quantification of the dimer factor identified here.

### 4.2.2 Dimers initiated by $NO_3$ radicals

Previous studies show that $NO_3$ oxidation of α-pinene, the most abundant monoterpene in Hyytiälä (Hakola et al., 2012), produces fairly little SOA mass (yields <4 %), while β-pinene shows yields of up to 53 % (Bonn and Moorgat, 2002;Nah et al., 2016). The $NO_3$+β-pinene reaction results in low volatile organic nitrate compounds with carboxylic acid, alcohol, and peroxide functional groups (Fry et al., 2014;Boyd et al., 2015), while $NO_3$+α-pinene reaction will typically lose the nitrate functional group and form oxidation products with high vapor pressures (Spittler et al., 2006;Perraud et al., 2010). Most monoterpene-derived HOM, including monomers, are low-volatile (Peräkylä et al., 2019) and thus a low SOA yield indicates a low HOM yield. Thus, while there are to our knowledge no laboratory studies on HOM formation from $NO_3$ oxidation of α-pinene, a low yield can be expected based on SOA studies.

As discussed in section 4.1.3, a dimer factor (R3F2_N2) was identified as a mix of ozonolysis and $NO_3$ oxidation processes, dominated by the organonitrate at 555 Th, $C_{20}H_{31}O_{10}NO_3 \cdot NO_3^-$. However, unlike the pure ozonolysis dimer factor which had a corresponding monomer factor (R = 0.86 between factor R2F4_N and R3 F2_N1), this $NO_3$-related dimer factor did not have an equivalent monomer factor. This suggests that the $NO_3$ oxidation of the monoterpene mixture in Hyytiälä does not by itself form much HOM, but in the presence of $RO_2$ from ozonolysis, the $RO_2$ from $NO_3$ oxidation can take part in HOM dimer formation. This further implies that, different from previous knowledge based on





single-oxidant experiments in chambers, $NO_3$ oxidation may have a larger impact on SOA formation
in the atmosphere where different oxidants exist concurrently. This highlights the need for future
laboratory studies to consider systems with multiple oxidants during monoterpene oxidation
experiments, to truly understand the role and contribution of different oxidants, and $NO_3$ in particular.

**5 Conclusions**
The recent development in mass spectrometry has greatly improved the detection of atmospheric
vapors and their oxidation products. Factor analysis, such as PMF, can reduce dimensionality of the
big datasets and extract factors relating to different atmospheric pathways/sources. Optimally, PMF
can link laboratory-generated spectra with ambient observations, significantly improving our
understanding of complicated atmospheric processes. However, one of PMF's basic assumptions is
that factor profiles remain constant in time, yet for atmospheric gas-phase species, varying sources,
reactions and sinks may violate this assumption. Some of these variations are likely not addressable
in the data analysis stage, but others may be. For example, molecules formed from the same source
can have different temporal behaviors due to varying volatilities, and thus condense at different rates.
Performing PMF separately with smaller ranges may circumvent this problem. By utilizing the newly
presented binPMF approach, more variables can be extracted from a narrow mass range compared to
traditional UMR PMF, while preserving more information in the spectrum.
We conducted separate binPMF analysis on three different sub-ranges to explore the potential benefits
of such an approach for producing more physically meaningful factors. We utilized ambient data
measured by CI-APi-TOF in a boreal forest environment, and selected sub-ranges from the mass
spectrum that roughly corresponded to regions where we would expect the molecules to have similar
volatilities and formation pathways. Selected ranges were Range 1 (250 – 300 Th), Range 2 (300 –
350 Th), and Range 3 (510 – 560 Th). binPMF was separately applied to these ranges, as well as to
the combination of all three for comparison.
The different sub-ranges produced some similar and some different factors. First of all, volatility of
species indeed affect the PMF results. We could clearly prove the benefit of sub-range binPMF using
the contamination factor as an example. We found that different compounds emitted from the same
source showed different temporal trends, likely due to differences in volatilities. This increased the
difficulties for PMF to separate this source in the combined data set, and the resolved profile was still
less accurate than in the analysis for the sub-ranges. We recommend that future studies of gas-phase
mass spectra should pay attention to this volatility effect on factor analysis.
Secondly, chemistry or sources contributing to the particular range can be better separated. Only the
binPMF analysis on Range 3, where HOM dimers are typically observed, resolved two nighttime



factors, characterized by monoterpene oxidation related to $NO_3$ and $O_3$ oxidation. The monoterpene
ozonolysis factor was separated from both Range 2 and 3, showing very good correlation between
the ranges and mutually verifying the results.
Thirdly, peaks with smaller signal intensities can be correctly assigned. The signal intensities between
different parts of the mass spectrum may vary by orders of magnitude. In the analysis of the combined
range, the results were almost completely controlled by the higher signals from Range 1 and 2. The
separate analysis on Range 3 allowed the low signals to provide important information, such as the
$NO_3$ oxidation process. In addition, running binPMF on different separate mass ranges also allows us
to compare the factors obtained from the different ranges and help to verify the results.
In addition, daytime dimer formation was identified, presumably initiated by $OH/O_3$ with a diurnal
peak at around noon, which may contribute to NPF in Hyytiälä. Also, based on the sub-range binPMF
analysis, we successfully separated $NO_3$–related dimers which did not have an equivalent monomer
factor. The $NO_3$ related factor was consistent with earlier observations (Yan et al., 2016), but would
not have been identified from this dataset without utilizing the different sub-ranges. In future
laboratory experiments, more complex oxidation systems may be useful in order to understand the
role $NO_3$ oxidation in SOA formation.
In summary, we identified several reasons to recommend PMF users to try running their analysis on
selected sub-ranges in addition to the whole spectra. Ultimately, the approach should be study-goal
dependent. In some cases the researcher wants a quick factor analysis to explore different features of
their data, while in others more accurate and quantitative separation of different sources and
atmospheric processes are needed. As binPMF and UMR-PMF both require very little data
preparation, we expect that it in most cases will be worth the time for the analyst to test how PMF
results look for a few selected sub-ranges of their mass spectra.

**Data availability.** The data used in this study are available from the first author upon request: please

contact Yanjun Zhang (yanjun.zhang@helsinki.fi).

Author contributions. ME and YZ designed the study. QZ and MR collected the data; data analysis
and manuscript writing were done by YZ. All coauthors discussed the results and commented the
manuscript.
**Competing interests.** The authors declare that they have no conflict of interest
**Acknowledgements.** We thank the tofTools team for providing tools for mass spectrometry data
analysis. The personnel of the Hyytiälä forestry field station are acknowledged for help during field
measurements.



**Financial support.** This research was supported by the European Research Council (Grant 638703-
COALA), the Academy of Finland (grants 317380 and 320094), and the Vilho, Yrjö and Kalle
Väisälä Foundation. K.R.D. acknowledges support by the Swiss National Science postdoc mobility
grant P2EZP2_181599.

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

A GLOBAL-MODEL OF NATURAL VOLATILE ORGANIC-COMPOUND EMISSIONS, Journal of
Geophysical Research-Atmospheres, 100, 8873-8892, 10.1029/94jd02950, 1995.
Hakola, H., Tarvainen, V., Bäck, J., Ranta, H., Bonn, B., Rinne, J., and Kulmala, M.: Seasonal variation of
mono- and sesquiterpene emission rates of Scots pine, Biogeosciences, 3, 93-101, 10.5194/bg-3-93-2006,
2006.

Hakola, H., Hellén, H., Hemmilä, M., Rinne, J., and Kulmala, M.: In situ measurements of volatile organic
compounds in a boreal forest, Atmos. Chem. Phys., 12, 11665-11678, 10.5194/acp-12-11665-2012, 2012.
Hari, P., and Kulmala, M.: Station for Measuring Ecosystem–Atmosphere Relations (SMEAR II), Boreal
Environment Research, 10, 315-322, 2005.
Huang, S., Rahn, K. A., and Arimoto, R.: Testing and optimizing two factor-analysis techniques on aerosol at
Narragansett, Rhode Island, Atmospheric Environment, 33, 2169-2185, https://doi.org/10.1016/S1352-
2310(98)00324-0, 1999.

Huffman, J. A., Docherty, K. S., Aiken, A. C., Cubison, M. J., Ulbrich, I. M., DeCarlo, P. F., Sueper, D., Jayne,
J. T., Worsnop, D. R., Ziemann, P. J., and Jimenez, J. L.: Chemically-resolved aerosol volatility
measurements from two megacity field studies, Atmos. Chem. Phys., 9, 7161-7182, 10.5194/acp-9-7161-
2009, 2009.

Jokinen, T., Sipilä, M., Junninen, H., Ehn, M., Lönn, G., Hakala, J., Petäjä, T., Mauldin Iii, R. L., Kulmala,
M., and Worsnop, D. R.: Atmospheric sulphuric acid and neutral cluster measurements using CI-APi-TOF,
Atmospheric Chemistry and Physics, 12, 4117-4125, 10.5194/acp-12-4117-2012, 2012.
Kirkby, J., Duplissy, J., Sengupta, K., Frege, C., Gordon, H., Williamson, C., Heinritzi, M., Simon, M., Yan,
C., Almeida, J., Troestl, J., Nieminen, T., Ortega, I. K., Wagner, R., Adamov, A., Amorim, A.,
Bernhammer, A.-K., Bianchi, F., Breitenlechner, M., Brilke, S., Chen, X., Craven, J., Dias, A., Ehrhart,
S., Flagan, R. C., Franchin, A., Fuchs, C., Guida, R., Hakala, J., Hoyle, C. R., Jokinen, T., Junninen, H.,
Kangasluoma, J., Kim, J., Krapf, M., Kuerten, A., Laaksonen, A., Lehtipalo, K., Makhmutov, V., Mathot,





S., Molteni, U., Onnela, A., Peraekylae, O., Piel, F., Petaejae, T., Praplan, A. P., Pringle, K., Rap, A.,
Richards, N. A. D., Riipinen, I., Rissanen, M. P., Rondo, L., Sarnela, N., Schobesberger, S., Scott, C. E.,
Seinfeld, J. H., Sipilae, M., Steiner, G., Stozhkov, Y., Stratmann, F., Tome, A., Virtanen, A., Vogel, A. L.,
Wagner, A. C., Wagner, P. E., Weingartner, E., Wimmer, D., Winkler, P. M., Ye, P., Zhang, X., Hansel,
A., Dommen, J., Donahue, N. M., Worsnop, D. R., Baltensperger, U., Kulmala, M., Carslaw, K. S., and
Curtius, J.: Ion-induced nucleation of pure biogenic particles, Nature, 533, 521-526, 10.1038/nature17953,
2016.
Kulmala, M., Kontkanen, J., Junninen, H., Lehtipalo, K., Manninen, H. E., Nieminen, T., Petäjä, T., Sipilä,
M., Schobesberger, S., Rantala, P., Franchin, A., Jokinen, T., Järvinen, E., Äijälä, M., Kangasluoma, J.,
Hakala, J., Aalto, P. P., Paasonen, P., Mikkilä, J., Vanhanen, J., Aalto, J., Hakola, H., Makkonen, U.,
Ruuskanen, T., Mauldin, R. L., Duplissy, J., Vehkamäki, H., Bäck, J., Kortelainen, A., Riipinen, I., Kurtén,
T., Johnston, M. V., Smith, J. N., Ehn, M., Mentel, T. F., Lehtinen, K. E. J., Laaksonen, A., Kerminen, V.-
M., and Worsnop, D. R.: Direct Observations of Atmospheric Aerosol Nucleation, 339, 943-946,
10.1126/science.1227385 %J Science, 2013.
Lamarque, J. F., Bond, T. C., Eyring, V., Granier, C., Heil, A., Klimont, Z., Lee, D., Liousse, C., Mieville, A.,
Owen, B., Schultz, M. G., Shindell, D., Smith, S. J., Stehfest, E., Van Aardenne, J., Cooper, O. R.,
Kainuma, M., Mahowald, N., McConnell, J. R., Naik, V., Riahi, K., and van Vuuren, D. P.: Historical
(1850–2000) gridded anthropogenic and biomass burning emissions of reactive gases and aerosols:
methodology and application, Atmos. Chem. Phys., 10, 7017-7039, 10.5194/acp-10-7017-2010, 2010.
Lee, B. H., Lopez-Hilfiker, F. D., Mohr, C., Kurtén, T., Worsnop, D. R., and Thornton, J. A.: An Iodide-
Adduct High-Resolution Time-of-Flight Chemical-Ionization Mass Spectrometer: Application to
Atmospheric Inorganic and Organic Compounds, Environmental Science & Technology, 48, 6309-6317,
10.1021/es500362a, 2014.
Lee, B. H., Lopez-Hilfiker, F. D., D'Ambro, E. L., Zhou, P., Boy, M., Petäjä, T., Hao, L., Virtanen, A., and
Thornton, J. A.: Semi-volatile and highly oxygenated gaseous and particulate organic compounds observed
above a boreal forest canopy, Atmos. Chem. Phys., 18, 11547-11562, 10.5194/acp-18-11547-2018, 2018.
Lehtipalo, K., Yan, C., Dada, L., Bianchi, F., Xiao, M., Wagner, R., Stolzenburg, D., Ahonen, L. R., Amorim,
A., Baccarini, A., Bauer, P. S., Baumgartner, B., Bergen, A., Bernhammer, A.-K., Breitenlechner, M.,
Brilke, S., Buchholz, A., Mazon, S. B., Chen, D., Chen, X., Dias, A., Dommen, J., Draper, D. C., Duplissy,
J., Ehn, M., Finkenzeller, H., Fischer, L., Frege, C., Fuchs, C., Garmash, O., Gordon, H., Hakala, J., He,
X., Heikkinen, L., Heinritzi, M., Helm, J. C., Hofbauer, V., Hoyle, C. R., Jokinen, T., Kangasluoma, J.,
Kerminen, V.-M., Kim, C., Kirkby, J., Kontkanen, J., Kürten, A., Lawler, M. J., Mai, H., Mathot, S.,
Mauldin, R. L., Molteni, U., Nichman, L., Nie, W., Nieminen, T., Ojdanic, A., Onnela, A., Passananti, M.,
Petäjä, T., Piel, F., Pospisilova, V., Quéléver, L. L. J., Rissanen, M. P., Rose, C., Sarnela, N., Schallhart,
S., Schuchmann, S., Sengupta, K., Simon, M., Sipilä, M., Tauber, C., Tomé, A., Tröstl, J., Väisänen, O.,
Vogel, A. L., Volkamer, R., Wagner, A. C., Wang, M., Weitz, L., Wimmer, D., Ye, P., Ylisirniö, A., Zha,
Q., Carslaw, K. S., Curtius, J., Donahue, N. M., Flagan, R. C., Hansel, A., Riipinen, I., Virtanen, A.,
Winkler, P. M., Baltensperger, U., Kulmala, M., and Worsnop, D. R.: Multicomponent new particle
formation from sulfuric acid, ammonia, and biogenic vapors, 4, eaau5363, 10.1126/sciadv.aau5363 %J
Science Advances, 2018.
Liebmann, J., Karu, E., Sobanski, N., Schuladen, J., Ehn, M., Schallhart, S., Quéléver, L., Hellen, H., Hakola,
H., Hoffmann, T., Williams, J., Fischer, H., Lelieveld, J., and Crowley, J. N.: Direct measurement of NO3
radical reactivity in a boreal forest, Atmos. Chem. Phys., 18, 3799-3815, 10.5194/acp-18-3799-2018, 2018.
Massoli, P., Stark, H., Canagaratna, M. R., Krechmer, J. E., Xu, L., Ng, N. L., Mauldin, R. L., Yan, C., Kimmel,
J., Misztal, P. K., Jimenez, J. L., Jayne, J. T., and Worsnop, D. R.: Ambient Measurements of Highly
Oxidized Gas-Phase Molecules during the Southern Oxidant and Aerosol Study (SOAS) 2013, ACS Earth
and Space Chemistry, 10.1021/acsearthspacechem.8b00028, 2018.
Mohr, C., Lopez-Hilfiker, F. D., Yli-Juuti, T., Heitto, A., Lutz, A., Hallquist, M., D'Ambro, E. L., Rissanen,
M. P., Hao, L., Schobesberger, S., Kulmala, M., Mauldin III, R. L., Makkonen, U., Sipilä, M., Petäjä, T.,
and Thornton, J. A.: Ambient observations of dimers from terpene oxidation in the gas phase: Implications
for new particle formation and growth, 44, 2958-2966, 10.1002/2017gl072718, 2017.
Nah, T., Sanchez, J., Boyd, C. M., and Ng, N. L.: Photochemical Aging of α-pinene and β-pinene Secondary
Organic Aerosol formed from Nitrate Radical Oxidation, Environmental Science & Technology, 50, 222-
231, 10.1021/acs.est.5b04594, 2016.



Orlando, J. J., and Tyndall, G. S.: Laboratory studies of organic peroxy radical chemistry: an overview with
emphasis on recent issues of atmospheric significance, J Chemical Society Reviews, 41, 6294-6317, 2012.
Paatero, P., and Tapper, U.: Positive matrix factorization: A non-negative factor model with optimal utilization
of error estimates of data values, Environmetrics, 5, 111-126, 1994.
Paatero, P.: Least squares formulation of robust non-negative factor analysis, Chemometrics and Intelligent
Laboratory Systems, 37, 23-35, https://doi.org/10.1016/S0169-7439(96)00044-5, 1997.
Paatero, P.: The Multilinear Engine—A Table-Driven, Least Squares Program for Solving Multilinear
Problems, Including the n-Way Parallel Factor Analysis Model, Journal of Computational and Graphical
Statistics, 8, 854-888, 10.1080/10618600.1999.10474853, 1999.
Paciga, A., Karnezi, E., Kostenidou, E., Hildebrandt, L., Psichoudaki, M., Engelhart, G. J., Lee, B. H., Crippa,
838       M., Prévôt, A. S. H., Baltensperger, U., and Pandis, S. N.: Volatility of organic aerosol and its components
in the megacity of Paris, Atmos. Chem. Phys., 16, 2013-2023, 10.5194/acp-16-2013-2016, 2016.
Paulson, S. E., and Orlando, J. J.: The reactions of ozone with alkenes: An important source of HOx in the
boundary layer, 23, 3727-3730, 10.1029/96gl03477, 1996.
Peräkylä, O., Riva, M., Heikkinen, L., Quéléver, L., Roldin, P., and Ehn, M.: Experimental investigation into
the volatilities of highly oxygenated organic molecules (HOM), Atmospheric Chemistry and Physics
Discussions, 2019, 1-28, 10.5194/acp-2019-620, 2019.
Perraud, V., Bruns, E. A., Ezell, M. J., Johnson, S. N., Greaves, J., and Finlayson-Pitts, B. J.: Identification of
Organic Nitrates in the NO3 Radical Initiated Oxidation of α-Pinene by Atmospheric Pressure Chemical
Ionization Mass Spectrometry, Environmental Science & Technology, 44, 5887-5893, 10.1021/es1005658,
2010.
Polissar, A. V., Hopke, P. K., Paatero, P., Malm, W. C., and Sisler, J. F.: Atmospheric aerosol over Alaska: 2.
Elemental composition and sources, Journal of Geophysical Research: Atmospheres, 103, 19045-19057,
1998.
Pope III, C. A., Ezzati, M., and Dockery, D. W.: Fine-particulate air pollution and life expectancy in the United
States, New England Journal of Medicine, 360, 376-386, 2009.
Riva, M., Rantala, P., Krechmer, J. E., Peräkylä, O., Zhang, Y., Heikkinen, L., Garmash, O., Yan, C., Kulmala,
855       M., Worsnop, D., and Ehn, M.: Evaluating the performance of five different chemical ionization
techniques for detecting gaseous oxygenated organic species, Atmospheric Measurement Techniques, 12,
2403-2421, 10.5194/amt-12-2403-2019, 2019.
Sekimoto, K., Koss, A. R., Gilman, J. B., Selimovic, V., Coggon, M. M., Zarzana, K. J., Yuan, B., Lerner, B.
859       M., Brown, S. S., Warneke, C., Yokelson, R. J., Roberts, J. M., and de Gouw, J.: High- and low-
temperature pyrolysis profiles describe volatile organic compound emissions from western US wildfire
fuels, Atmos. Chem. Phys., 18, 9263-9281, 10.5194/acp-18-9263-2018, 2018.
Shiraiwa, M., Ueda, K., Pozzer, A., Lammel, G., Kampf, C. J., Fushimi, A., Enami, S., Arangio, A. M.,
Fröhlich-Nowoisky, J., Fujitani, Y., Furuyama, A., Lakey, P. S. J., Lelieveld, J., Lucas, K., Morino, Y.,
Pöschl, U., Takahama, S., Takami, A., Tong, H., Weber, B., Yoshino, A., and Sato, K.: Aerosol Health
Effects from Molecular to Global Scales, Environmental Science & Technology, 51, 13545-13567,
10.1021/acs.est.7b04417, 2017.
Song, Y., Shao, M., Liu, Y., Lu, S., Kuster, W., Goldan, P., and Xie, S.: Source apportionment of ambient
volatile organic compounds in Beijing, Environmental science & technology, 41, 4348-4353, 2007.
Spittler, M., Barnes, I., Bejan, I., Brockmann, K. J., Benter, T., and Wirtz, K.: Reactions of NO3 radicals with
limonene and α-pinene: Product and SOA formation, Atmospheric Environment, 40, 116-127,
https://doi.org/10.1016/j.atmosenv.2005.09.093, 2006.
Stocker, T., Qin, D., Plattner, G., Tignor, M., Allen, S., Boschung, J., Nauels, A., Xia, Y., Bex, V., and Midgley,
P.: IPCC, 2013: Climate Change 2013: The Physical Science Basis. Contribution of Working Group I to
the Fifth Assessment Report of the Intergovernmental Panel on Climate Change, 1535 pp, in, Cambridge
Univ. Press, Cambridge, UK, and New York, 2013.
Troestl, J., Chuang, W. K., Gordon, H., Heinritzi, M., Yan, C., Molteni, U., Ahlm, L., Frege, C., Bianchi, F.,
Wagner, R., Simon, M., Lehtipalo, K., Williamson, C., Craven, J. S., Duplissy, J., Adamov, A., Almeida,
878       J., Bernhammer, A.-K., Breitenlechner, M., Brilke, S., Dias, A., Ehrhart, S., Flagan, R. C., Franchin, A.,
Fuchs, C., Guida, R., Gysel, M., Hansel, A., Hoyle, C. R., Jokinen, T., Junninen, H., Kangasluoma, J.,
Keskinen, H., Kim, J., Krapf, M., Kuerten, A., Laaksonen, A., Lawler, M., Leiminger, M., Mathot, S.,
Moehler, O., Nieminen, T., Onnela, A., Petaejae, T., Piel, F. M., Miettinen, P., Rissanen, M. P., Rondo,
882       L., Sarnela, N., Schobesberger, S., Sengupta, K., Sipila, M., Smith, J. N., Steiner, G., Tome, A., Virtanen,



A., Wagner, A. C., Weingartner, E., Wimmer, D., Winkler, P. M., Ye, P., Carslaw, K. S., Curtius, J.,
Dommen, J., Kirkby, J., Kulmala, M., Riipinen, I., Worsnop, D. R., Donahue, N. M., and Baltensperger,
U.: The role of low-volatility organic compounds in initial particle growth in the atmosphere, Nature, 533,
527-531, 10.1038/nature18271, 2016.
Ulbrich, I. M., Canagaratna, M. R., Zhang, Q., Worsnop, D. R., and Jimenez, J. L.: Interpretation of organic
components from Positive Matrix Factorization of aerosol mass spectrometric data, Atmos. Chem. Phys.,
9, 2891-2918, 10.5194/acp-9-2891-2009, 2009.
Yan, C., Nie, W., Aijala, M., Rissanen, M. P., Canagaratna, M. R., Massoli, P., Junninen, H., Jokinen, T.,
Sarnela, N., Hame, S. A. K., Schobesberger, S., Canonaco, F., Yao, L., Prevot, A. S. H., Petaja, T., Kulmala,
M., Sipila, M., Worsnop, D. R., and Ehn, M.: Source characterization of highly oxidized multifunctional
compounds in a boreal forest environment using positive matrix factorization, Atmospheric Chemistry and
Physics, 16, 12715-12731, 10.5194/acp-16-12715-2016, 2016.
Zha, Q., Yan, C., Junninen, H., Riva, M., Sarnela, N., Aalto, J., Quéléver, L., Schallhart, S., Dada, L.,
Heikkinen, L., Peräkylä, O., Zou, J., Rose, C., Wang, Y., Mammarella, I., Katul, G., Vesala, T., Worsnop,
D. R., Kulmala, M., Petäjä, T., Bianchi, F., and Ehn, M.: Vertical characterization of highly oxygenated
molecules (HOMs) below and above a boreal forest canopy, Atmos. Chem. Phys., 18, 17437-17450,
10.5194/acp-18-17437-2018, 2018.
Zhang, Q., Jimenez, J. L., Canagaratna, M. R., Ulbrich, I. M., Ng, N. L., Worsnop, D. R., and Sun, Y.:
Understanding atmospheric organic aerosols via factor analysis of aerosol mass spectrometry: a review,
Analytical and Bioanalytical Chemistry, 401, 3045-3067, 10.1007/s00216-011-5355-y, 2011.
Zhang, Y., Lin, Y., Cai, J., Liu, Y., Hong, L., Qin, M., Zhao, Y., Ma, J., Wang, X., and Zhu, T.: Atmospheric
PAHs in North China: spatial distribution and sources, Science of the Total Environment, 565, 994-1000,
2016.
Zhang, Y., Cai, J., Wang, S., He, K., and Zheng, M.: Review of receptor-based source apportionment research
of fine particulate matter and its challenges in China, Science of the Total Environment, 586, 917-929,
2017.
Zhang, Y., Peräkylä, O., Yan, C., Heikkinen, L., Äijälä, M., Daellenbach, K. R., Zha, Q., Riva, M., Garmash,
O., Junninen, H., Paatero, P., Worsnop, D., and Ehn, M.: A novel approach for simple statistical analysis
of high-resolution mass spectra, Atmospheric Measurement Techniques, 12, 3761-3776, 10.5194/amt-12-
912 3761-2019, 2019.