# Peer review of "Insights on Atmospheric Oxidation Processes by Performing Factor Analyses on"

_Atmospheric Chemistry and Physics, 2019_

## Referee Comment (RC1) · Anonymous Referee #1 · 27 Dec 2019

This manuscript promotes two new approaches to analyze complex mass spectra (here of highly oxygenated molecules (HOM)) with the goal to extract as much as possible information from the whole mass spectroscopic information: binwise PMF and coordinated PMF analysis in selected mass ranges. The authors suggest to select certain mass ranges for analysis according to expected time scales of production processes and sink processes, whereby they have condensation loss as major sink in mind. The approach is exercised at an ambient data set, measured by NO3-CI_API-TOF in September 2016 at the SMEAR II station in Finland. The manuscript is very well written, it is informative and very interesting to read. It discusses the limits (and strengths) of PMF analysis of atmospheric observations in the context of the variabil-

ity of production and sinks processes of gaseous compounds in the atmosphere. It also points out two new observations - day time dimers and night time dimer nitrates-, which need mechanistic explanations. The focus of the paper is on the methodology, although along its development it reveals insight into HOM formation processes. I suggest highlighting the latter already more in the Discussion sections and summarizing it again in Atmospheric Insights section. I also suggest shortening the Contaminant Factor section and place some text from here into the supplement. See also comments. In the larger parts I see the manuscript - even in its given form - as an original and important contribution of general interest to atmospheric scientists, as nowadays in many atmospheric fields research is based on high resolution mass spectrometry. I suggest publishing the manuscript in ACP after the authors have addressed the (minor) comments below.

Comments

Abstract, line 36: Don't present your new findings as appendix. "As two mayor insights of our analysis scheme, we identified daytime dimer formation... We separated dimer formation by NO3 oxidation. ... "

Introduction, line 87: Could the authors comment on the role of chemical losses compared to condensation losses onto particles. Couldn't chemical losses, e.g. by oxidation of HOM by OH, enhance the window of sink time scales?

Introduction, line 93: Peräkylä et al., 2019 is not a suited reference for such a general statement.

Introduction, line 121: The impact of the oxidants is different at different times of the day. That should limit important formation pathways to less than 6.

Introduction, line 126: RO2 + RO2 also lead to monomer termination products. Thus, the statement is not valid in such generality. Maybe on should modify the sentence: ....in monomer products (not terminated by RO2), dependent on only one oxidant. ...

Introduction, line 145: It would be wishful to refer also already here to the new atmospheric information not only to "meaningful factors". In the sense of "we will show that we were able to separate process x from process y".

Result, line 295: This sentence is hard to understand, please, split and reformulate.

Discussion, line 448: hard to understand and possibly a verb missing. Please, reformulate.

Discussion, line 468 and 474: Did you search for specific marker ions (odd mass) in the monomer range? Maybe, singular nitrates are formed quite efficiently and the corresponding nitrate peroxy radicals could be involved in the dimer formation. Could NO3 radicals attack the dimers the dimer?

Discussion, line 485-548: I find that whole section too lengthy. I agree that the authors performed a smart analysis to find out why contamination factors do correlate or not. However, is this argumentation really needed to demonstrate that different loss rates lead to different time profiles, thus attribution to different factors, if the source is the same?

Insofar I find the jump from the contamination analysis to ambient data in line 541 somewhat disturbing. At least a new paragraph should start here.

I suggest to place the text of contamination analysis in large parts into the supplement. In the manuscript I would just state that a detailed analysis explained why contamination factors do not correlate and refer to the supplement. The space saved could be used to more underline the atmospheric findings somewhat more (all over the manuscript and in the Atmospheric Insights section).

Atmospheric Insights, line 550: "While the previous section discussed several findings with atmospheric implications,.." I suggest to sample and to discuss at this point the insights into the HOM formation processes mentioned in all the discussion sections. And maybe elaborate the two new findings somewhat more.

Conclusion, line 633: As mentioned before, highlight the new atmospheric findings here. Prevent presenting it as an appendix to your methodological approach.

Minor

Figure captions are not separated well from running text.

line 119: any instead of many ?

line 468: Please, replace "this factor" by the name of the factor "factor R2F4_N" for faster readability, because there was more than one factor listed in the previous sentence.

---

## Referee Comment (RC2) · Anonymous Referee #2 · 10 Jan 2020

This manuscript presents binPFM (Positive matrix factorization) analysis results of sub-ranges of mass spectra and combined ranges of mass spectra, respectively. The authors compared the results from three sub-ranges and the combined three, and concluded that the PFM results depended on the volatility of the species that is assumed to be identical among species, the chemistry or source that contributes to a particulate range of species, and the relative abundance of different species. The authors also discussed the potential formation mechanisms of observed species, especially dimers formed from peroxy radicals. Generally this is a very interesting study that clearly shows the potential issue when applying the PMF methods to measurements of volatile organic compound with different volatilities, which is of interest to the atmospheric chemistry community. On the other hand, the manuscript is a little bit too technical for Atmospheric chemistry and physics, but can be revised to fit. I would recommend publication of this manuscript after the following concerns have been addressed.

1. Overall, this manuscript focuses too much on the method itself but does not put enough weight on the science they have obtained by analyzing the dataset. The current organization is more like an AMT paper instead of an ACP one. The authors are advised to move a certain fraction of the technical part, e.g., the contamination session, into the supplement and expand the scientific findings.

2. The texts in the conclusion part are quite redundant and just a repeat of the issues with applying traditional PMF to CIMS data, especially in the first two paragraphs. This part certainly can be rewritten to be more concise and to deliver key conclusions only.

3. (Line 121), "six different pathways" would not be the best word, since OH and NO3 chemistry would not generally happen at the same time. Although OH radicals can be generated from pinene+ ozone chemistry at night, the chances for cross reactions of dimers between peroxy radicals formed from OH chemistry and those from nitrate chemistry are just low, in my mind.

4. (Line 220-232), a couple of statements should be clarified. There is a statement of a bin width of 0.02 Th (Line 221). On the other hand, authors state "25 bins per unit mass" for Ranges 1 and 2 and "30 bins per unit mass" for Range 3. What caused the difference? Also, I assume that a larger range of signal region for Range 3 in further analysis was due to a worse shift in mass-to-charge? Lastly, what is the setup for the combined range analysis?

5. (Sessions 3.2-3.5), since the authors started from two factor analysis to more factors, point out the sequence of factors presented does not necessarily correspond that in the figures in each session.

6. (Figures 2-5), state the time period for the factor contribution.

7. (Line 361-362, 455-456), isn't it true that the ultimate source of NO during daytime is still emission?

8. (Line 435), the termination of one peroxy radicals with another does not necessarily have to lead to the formation of dimers.

9. (Line 497-499), rephrase the sentence.

10. (Session 4.2.1) Although there might not be a daytime factor previously, it is not surprising that dimers are formed in the day. NO channel competes with the self reactions of peroxy radicals but which level of NO will really dominate is still an open question.

---

## Author Comment (AC1) · 6 Mar 2020

The comment was uploaded in the form of a supplement.

Please also note the supplement to this comment:
https://www.atmos-chem-phys-discuss.net/acp-2019-838/acp-2019-838-AC1-supplement.pdf

———————————————

---

## Author Comment (AC2) · 6 Mar 2020

**Responses to Reviewers' Comments**

We would like to thank the reviewers for their valuable and positive feedback/comments, which helped us to improve the manuscript.

Referee comments are given in black, and our point-to-point responses are in green. Changes made to the manuscript are marked in underlined green. The line number referred here is for the new revised manuscript.

**Anonymous Referee #1**

This manuscript promotes two new approaches to analyze complex mass spectra (here of highly oxygenated molecules (HOM)) with the goal to extract as much as possible information from the whole mass spectroscopic information: binwise PMF and coordinated PMF analysis in selected mass ranges. The authors suggest to select certain mass ranges for analysis according to expected time scales of production processes and sink processes, whereby they have condensation loss as major sink in mind. The approach is exercised at an ambient data set, measured by NO3-CI_APITOF in September 2016 at the SMEAR II station in Finland. The manuscript is very well written, it is informative and very interesting to read. It discusses the limits (and strengths) of PMF analysis of atmospheric observations in the context of the variability of production and sinks processes of gaseous compounds in the atmosphere. It also points out two new observations - day time dimers and night time dimer nitrates-, which need mechanistic explanations. The focus of the paper is on the methodology, although along its development it reveals insight into HOM formation processes. I suggest highlighting the latter already more in the Discussion sections and summarizing it again in Atmospheric Insights section. I also suggest shortening the Contaminant Factor section and place some text from here into the supplement. See also comments. In the larger parts I see the manuscript - even in its given form - as an original and important contribution of general interest to atmospheric scientists, as nowadays in many atmospheric fields research is based on high resolution mass spectrometry. I suggest publishing the manuscript in ACP after the authors have addressed the (minor) comments below.

Comments

Abstract, line 36: Don't present your new findings as appendix. "As two mayor insights of our analysis scheme, we identified daytime dimer formation. . . We separated dimer formation by NO3 oxidation.... "

*Response 1.1:* We agree with the reviewer. We have revised this part from "In addition, daytime dimer formation (diurnal peak around noon) was identified, which may contribute to NPF in Hyytiälä. Also, dimers from $NO_3$ oxidation were separated by the sub-range binPMF, which would not be identified otherwise." to

"As two major insights from our study, we identified daytime dimer formation (diurnal peak around noon) which may contribute to NPF in Hyytiälä, as well as dimers from $NO_3$ oxidation process." in Line 36-38.

Introduction, line 87: Could the authors comment on the role of chemical losses compared to condensation losses onto particles. Couldn't chemical losses, e.g. by oxidation of HOM by OH, enhance the window of sink time scales?

*Response 1.2:* Both Peräkylä et al. (2020) and Bianchi et al. (2019) assess the impact of oxidation on the HOM lifetime, and find that it's negligible under typical conditions. Condensation dominates, even when assuming collision limited reaction with OH radicals. However, for compounds of higher volatility, such as IVOC and SVOC, oxidation could reduce their lifetime. We added the following explanation in Line 87:

"……which may affect the factor analysis. For compounds of low volatility, such as many HOM, the main atmospheric loss process is typically condensation onto aerosol particles, with chemical sink being negligible (Bianchi et al., 2019). If, on the other hand, a compound does not irreversibly condense, oxidation reactions can also affect its lifetime."

Introduction, line 93: Peräkylä et al., 2019 is not a suited reference for such a general statement.

*Response 1.3:* Ok, we removed this reference from here.

Introduction, line 121: The impact of the oxidants is different at different times of the day. That should limit important formation pathways to less than 6.

*Response 1.4:* We agree with the reviewer that different oxidants have their major impacts during different times of a day. Our initial goal here was just to point out there can be at most six pathways to form dimers, if considering the same precursor, formed from the same or different oxidants ($O_3$, OH, $NO_3$), so as to remind the readers that molecules from different ranges may have different formation pathways. To minimize the misunderstanding, we modified the sentence into

"……. $RO_2+RO_2$ reactions (Berndt et al., 2018a;Berndt et al., 2018b). This also means that there can be several different pathways to form dimers from the same precursor VOC, by combining $RO_2$ formed from the same or different oxidants. As an example of the latter ……" in Line 124.

Introduction, line 126: RO2 + RO2 also lead to monomer termination products. Thus, the statement is not valid in such generality. Maybe on should modify the sentence: . . ..in monomer products (not terminated by RO2), dependent on only one oxidant. . ..

*Response 1.5:* For $RO_2$ + $R'O_2$ reactions, as reviewed by Orlando and Tyndall (2012) and (Bianchi et al., 2019), there are two main direct $RO_2$ termination reactions:

$RO_2 + R'O_2 \rightarrow ROH + R'C=O + O_2$     (a)

$RO_2 + R'O_2 \rightarrow ROOR' + O_2$   (b)

Reaction channel (a) will lead to monomer termination products, and (b) will lead to dimer products. It is true that $RO_2 + RO_2$ forms also monomer products. However, what we specifically meant, is that in this case, the formed monomer does not care about the identity of the other $RO_2$ taking part in the reaction, other than potentially about whether a carbonyl or an alcohol will be formed (Orlando and Tyndall, 2012). Thus, to a first approximation, in the $RO_2+RO_2$ terminated

monomer channel, any $RO_2$ will do as the terminator. This is in contrast with the dimer case, where the identity of both $RO_2$ will impact the formed dimer. To make it clear, we replaced the sentences in Line 128-133 "Such a molecule will not have a direct equivalent in any of the monomer products, dependent on only one oxidant, which again may complicate the separation of such factors by PMF, if the entire spectrum is analyzed once. However, if separating the monomer and dimer products before PMF analysis, separation of different formation pathways can potentially become simpler.",

with "Such a molecule will not have a direct equivalent in any of the monomer products: even though monomers can form from $RO_2$ + $R'O_2$ reactions, the products from $RO_2$ are not dependent on the source of the $R'O_2$. This may complicate the identification of certain dimer factors by PMF if the entire spectrum is analyzed at once, and therefore separation of the monomer and dimer products before the PMF analysis could be advantageous."

Introduction, line 145: It would be wishful to refer also already here to the new atmospheric information not only to "meaningful factors". In the sense of "we will show that we were able to separate process x from process y".

*Response 1.6:* We have revised this part in Line 145 from "…… run on the combined ranges. We found that more meaningful factors are separated from our dataset by utilizing the sub-ranges, and believe that this study will provide new perspectives for future studies …..." to

"…… run on the combined ranges. We found that dimers generated during daytime and dimers initiated by $NO_3$ oxidation can be separated from our dataset by utilizing the sub-ranges, but not with the full range. We believe that this study will provide new perspectives for future studies …..."

Result, line 295: This sentence is hard to understand, please, split and reformulate.

*Response 1.7:* To make it clearer, we have changed the sentence from "…… Factors 1-3 are all daytime factors, while Factor 4 has a sawtooth shape, which is caused by contamination, mainly by perfluorinated acids, of the inlet's automated zeroing every three hours during the measurements (Zhang et al., 2019)." to

"…… Factors 1-3 are all daytime factors, while Factor 4 has no clear diurnal cycle, but a distinct sawtooth shape. Factor 4 comes from a contamination of perfluorinated acids, from the inlet's automated zeroing every three hours (Zhang et al., 2019)." in Line 313-315.

Discussion, line 448: hard to understand and possibly a verb missing. Please, reformulate.

*Response 1.8:* To make it easier to understand, we have changed the sentence from "…… (R3F1_D) basically has no obvious markers in the profile, and as mentioned above, up to ten factors, there would only be more factors fragmented from the previous factor, with similar spectral profiles, but showed different profile pattern with 510 – 560 Th in RCF2_D2 in Range Combined. The factorization of Range Combined was mainly controlled by Range 1 and 2 due to high signals, and the signals in Range 3 are forced to be distributed according to the time series determined by Ranges 1 and 2. Ultimately, this will lead to …..." to

"…… (R3F1_D) has no obvious markers in the profile. With the increase of factor number (up to ten factors), no clearly new factors were separated in Range 3, but instead the previously

separated factors were seen to split into several factors. However, the spectral pattern in R3F1_D is different from that in the mass range of 510 – 560 Th in RCF2_D2. The factorization of Range Combined was mainly controlled by low masses due to their high signals. The signals at high masses were forced to be distributed according to the time series determined by small masses. Ultimately, this will lead to …...." in Line 476-482.

Discussion, line 468 and 474: Did you search for specific marker ions (odd mass) in the monomer range? Maybe, singular nitrates are formed quite efficiently and the corresponding nitrate peroxy radicals could be involved in the dimer formation. Could NO3 radicals attack the dimers the dimer?

***Response 1.9:*** **(1)** As the reviewer suggests, the possible monomers initiated by $NO_3$ oxidation would have odd integer masses, as listed in Table R1.

Table R1 potential monomers initiated by $NO_3$ oxidation and their corresponding integer masses

| Number of O atom (x) | $C_{10}H_{15}NO_x NO_3-$ | $C_{10}H_{17}NO_x NO_3-$ |
|---|---|---|
| 1 | 227 | 229 |
| 2 | 243 | 245 |
| 3 | 259 | 261 |
| 4 | 275 | 277 |
| 5 | 291 | 293 |
| 6 | 307 | 309 |
| 7 | 323 | 325 |
| 8 | 339 | 341 |
| 9 | 355 | 357 |

In the monomer range (Range 2), the only night factor (as shown below in Figure R1) is R2F4_N. 325 Th is the highest odd mass in Range 2, with no other significant odd-mass markers. We can do a high-resolution fit to 325 Th, with center mass of 325.06 Th and resolution of 3577 Th/Th, which is quite close to the instrument resolution, ~4000 Th/Th. There can be mainly two candidates for 325 Th, the radical $C_{10}H_{15}O_8 NO_3^-$ (with exact mass of 325.0651 Th) and monomer from $NO_3$ oxidation $C_{10}H_{17}NO_7 NO_3^-$ (with exact mass of 325.0889 Th). In our case, this 325 Th is more likely the radical. In addition, R2F4_N shows significantly higher correlation with ozonolysis dimers (R3F2_N1) ($R^2 = 0.75$) compared to the $NO_3$ oxidation dimer (R3F3_N2) ($R^2 = 0.27$). A similar result is also found in the combined range, as shown in Figure R2. We thus conclude that any potential monomer nitrates are minor compared to the non-nitrate monomers, and an instrument with much higher resolution, e.g. the CI-Orbitrap (Riva et al., 2019), would be needed to unambiguously identify such compounds.

[Figure]

Figure R1 Spectral profile of R2F4_N in fraction, with the main m/z marked

[Figure]

Figure R2 Spectral profile of RCF4_N in fraction, with the main m/z marked in the monomer range

The detection of nitrate dimers of course mean that $NO_3$-initiated radicals do form efficiently, and participate in the dimer formation by providing the $RO_2$ supply. We only argue that the potential to undergo autoxidation, and thus form HOM monomers, is limited for the monoterpenes during these measurements. As even less oxidized dimers condense very efficiently, they will have very limited (see also *Response 1.2*) time to react with $NO_3$ radicals, meaning that it's unlikely that the observed nitrate dimers would be formed from a reaction between nitrate radicals and non-nitrate dimers.

Discussion, line 485-548: I find that whole section too lengthy. I agree that the authors performed a smart analysis to find out why contamination factors do correlate or not. However, is this argumentation really needed to demonstrate that different loss rates lead to different time profiles, thus attribution to different factors, if the source is the same? Insofar I find the jump from the

contamination analysis to ambient data in line 541 somewhat disturbing. At least a new paragraph should start here. I suggest to place the text of contamination analysis in large parts into the supplement. In the manuscript I would just state that a detailed analysis explained why contamination factors do not correlate and refer to the supplement. The space saved could be used to more underline the atmospheric findings somewhat more (all over the manuscript and in the Atmospheric Insights section).

*Response 1.10:* We agree with the reviewer, and have moved most of the text of this part to supplement. Only the results, discussions of low correlations between different contamination compounds, and indications of volatility effect on factor analysis were kept.

Atmospheric Insights, line 550: "While the previous section discussed several findings with atmospheric implications,.." I suggest to sample and to discuss at this point the insights into the HOM formation processes mentioned in all the discussion sections. And maybe elaborate the two new findings somewhat more.

*Response 1.11:* Thanks for the comments. **(1)** For the "4. Discussion" part, we adjusted and revised the structure of this part, and rearranged and added the relative contents in different sections, respectively. The previous structure in the "4. Discussion" was:

"4. Discussion
 4.1 Comparison of different ranges
 4.1.1 Time series correlation
 4.1.2 Daytime factor comparison
 4.1.3 Nighttime factor comparison
 4.1.4 Contamination factor
 4.2 Atmospheric insights
 4.2.1 Daytime dimer formation
 4.2.2 Dimers initiated by $NO_3$ radicals",

while the new adjusted structure is

"4. Discussion
 4.1 Time series correlation
 4.2 Daytime processes
 4.2.1 Factor comparison
 4.2.2 Daytime dimer formation
 4.3 Nighttime processes
 4.3.1 Factor comparison
 4.3.2 Dimers initiated by $NO_3$ radicals
 4.4 Fluorinated compounds
 4.5 Atmospheric insights".

**(2)** Now the section "4.5 Atmospheric insights" was simplified and mainly summarized the key atmospheric insights which have been discussed in more details in sections 4.1-4.3.

[revised manuscript text omitted]

Conclusion, line 633: As mentioned before, highlight the new atmospheric findings here. Prevent presenting it as an appendix to your methodological approach. Minor Figure captions are not separated well from running text.

*Response 1.12:* **(1)** For the conclusion, as both reviewer suggested, changes have been made to simplify the conclusion and highlight the new findings. The new revised conclusion is as follows:

"The recent development in mass spectrometry, combined with factor analysis such as PMF, has greatly improved our understanding of complicated atmospheric processes and sources. However, one of PMF's basic assumptions is that factor profiles remain constant in time, yet for atmospheric gas-phase species, reactions and sinks may violate this assumption. In this study, we conducted separate binPMF analysis on three different sub-ranges to explore the potential benefits of such an approach for producing more physically meaningful factors.

With binPMF applied on sub-ranges, our study identified daytime dimers, presumably initiated by OH/$O_3$ with a diurnal peak at around noon, which may contribute to NPF in Hyytiälä. Also, based on the sub-range binPMF analysis, we successfully separated $NO_3$–related dimers which

did not have a corresponding monomer factor. The NO$_3$-related factor was consistent with earlier observations (Yan et al., 2016), but would not have been identified from this dataset without utilizing the different sub-ranges. In future laboratory experiments, more complex oxidation systems may be useful in order to understand the role of NO$_3$ oxidation in SOA formation. Apart from these two findings, we also find other benefits by applying binPMF on sub-ranges of the mass spectra.

First, volatility affects the PMF results. Different compounds emitted from the same source showed different temporal trends, likely due to differences in volatilities. This increased the difficulties for PMF to separate this source in the combined data set, and the resolved profile was less accurate than that of the sub-ranges. Future studies of gas-phase mass spectra should pay attention to this volatility effect on factor analysis.

Secondly, chemistry or sources contributing to the particular range can be better separated. Only the binPMF analysis on Range 3, where HOM dimers are typically observed, resolved two nighttime factors, characterized by monoterpene oxidation related to NO$_3$ and O$_3$ oxidation.

Thirdly, peaks with smaller signal intensities can be correctly assigned. The signal intensities between different parts of the mass spectrum may vary by orders of magnitude. In the combined case, the results were almost completely controlled by the higher signals from smaller masses. The separate analysis on Range 3 allowed the low signals to provide important information. In addition, running binPMF on different separate mass ranges also allows us to compare the factors obtained from the different ranges and help to verify the results."

**(2)** Minor figure captions are modified into smaller font size, so as to be better separated from the main text.

line 119: any instead of many ?

*Response 1.13:* Yes, changed.

line 468: Please, replace "this factor" by the name of the factor "factor R2F4_N" for faster readability, because there was more than one factor listed in the previous sentence.

*Response 1.14:* Yes, changed.

**Anonymous Referee #2**

This manuscript presents binPFM (Positive matrix factorization) analysis results of subranges of mass spectra and combined ranges of mass spectra, respectively. The authors compared the results from three sub-ranges and the combined three, and concluded that the PFM results depended on the volatility of the species that is assumed to be identical among species, the chemistry or source that contributes to a particulate range of species, and the relative abundance of different species. The authors also discussed the potential formation mechanisms of observed species, especially dimers formed from peroxy radicals. Generally this is a very interesting study that clearly shows the potential issue when applying the PMF methods to measurements of volatile organic compound with different volatilities, which is of interest to the atmospheric chemistry community. On the other hand, the manuscript is a little bit too technical for Atmospheric chemistry and physics, but can be revised to

fit. I would recommend publication of this manuscript after the following concerns have been addressed.

1. Overall, this manuscript focuses too much on the method itself but does not put enough weight on the science they have obtained by analyzing the dataset. The current organization is more like an AMT paper instead of an ACP one. The authors are advised to move a certain fraction of the technical part, e.g., the contamination session, into the supplement and expand the scientific findings.

*Response 2.1*: We agree with both reviewers, and have moved most of the text on contamination factor to the supplement. Quite a lot of revisions and adjustments in the part of "4. Discussion" were made to expand the scientific discussions and findings. Details are referred to *Response 1.11* of *Reviewer #1.*

2. The texts in the conclusion part are quite redundant and just a repeat of the issues with applying traditional PMF to CIMS data, especially in the first two paragraphs. This part certainly can be rewritten to be more concise and to deliver key conclusions only.

*Response 2.2:* In the conclusion part, we combined the first two paragraphs into one short paragraph, and also largely simplified other conclusions, as well as highlighting more the new atmospheric findings, as in *Response 1.12*.

3. (Line 121), "six different pathways" would not be the best word, since OH and NO3 chemistry would not generally happen at the same time. Although OH radicals can be generated from pinene+ ozone chemistry at night, the chances for cross reactions of dimers between peroxy radicals formed from OH chemistry and those from nitrate chemistry are just low, in my mind.

*Response 2.3*: We agree with both reviewers. To eliminate misunderstandings, we have changed the sentence. Detail are referred to the *Response 1.4.*

4. (Line 220-232), a couple of statements should be clarified. There is a statement of a bin width of 0.02 Th (Line 221). On the other hand, authors state "25 bins per unit mass" for Ranges 1 and 2 and "30 bins per unit mass" for Range 3. What caused the difference? Also, I assume that a larger range of signal region for Range 3 in further analysis was due to a worse shift in mass-to-charge? Lastly, what is the setup for the combined range analysis?

*Response 2.4:* Thanks for the comments. (**1**) The bin width in this study is 0.02 Th. To eliminate unnecessary computation of masses without any signal, only masses in the signal region (regions containing meaningful signals) were binned. The peaks get progressively wider with increasing m/z ratio, so at the higher masses of Range 3 we used a wider window for the signal. In this study, the signal region for Range 1 and 2 is between $N - 0.2$ and $N + 0.3$ Th, at integer mass N, and $N - 0.2$ and $N + 0.4$ Th for Range 3 (Figure R4). So the bins per unit mass for Ranges 1 and 2 is 0.5 Th / 0.02 Th = 25, and for 0.6 Th / 0.02 Th = 30 bins.

[Figure]

Figure R4 Schematic diagram of data matrix binning process for binPMF analysis. In this study, the bin width is 0.02 Th. For Ranges 1 & 2, the signal region for binning is [N - 0.2, N + 0.3], and for Range 3 is [N - 0.2, N + 0.4].

 **(2)** For the combined range in this analysis, we just combined the mass spectra in the above three ranges, i.e. the three datasets in Ranges 1-3 were combined together to construct combined range. Thus, the highest masses have more bins per integer mass than the mid- and low-mass ranges.

To clarity, we added a statement in Line 240-243:

"To avoid unnecessary computation, only signal regions with meaningful signals in the mass spectra were binned (Zhang et al., 2019). For a nominal mass $N$, the signal region included in further analyses was between $N$-0.2 Th and $N$+0.3 Th for Range 1 and 2, and between $N$-0.2 Th and $N$+0.4 Th for Range 3. The wider signal regions in Range 3 is due to wider peaks at higher masses. The data were averaged into 1-h time resolution and in total we had 384 time points in the data matrix."

5. (Sessions 3.2-3.5), since the authors started from two factor analysis to more factors, point out the sequence of factors presented does not necessarily correspond that in the figures in each session.

*Response 2.5:* We added two sentences in section 3.1 to clarify this.

"…… separately for each Range (sections 3.2 – 3.5). It is worth noting that the factor order in factor evolution does not necessarily correspond to that of the final results. The factor orders displayed in Figures 2-5 have been modified for further comparison between different ranges. More detailed …..." in Line 288-290.

6. (Figures 2-5), state the time period for the factor contribution.

*Response 2.6:* The factor contribution is an average from the whole measurement period. For the text, the relative sentence was modified in section 3.2-3.5, respectively, to make this clear.

"…… for further discussion, and Figure 2 shows the result of Range 1, with spectral profile, time series, diurnal cycle and averaged factor contribution during the campaign……." in Line 313.

"…… Figure 3 shows four-factor result of Range 2, with spectral profile, time series, diurnal cycle and averaged factor contribution during the campaign." in Line 348-350.

"……goal in this study. Figure 5 shows the four-factor result of Range Combined, with spectral profile, time series, diurnal cycle and averaged factor contribution during the campaign. The signals in……" in Line 394-395.

Figure caption in Figure 2-5 were also modified, from "(2) factor contribution" to "(2) averaged factor contribution during the campaign".

7. (Line 361-362, 455-456), isn't it true that the ultimate source of NO during daytime is still emission?

***Response 2.7***: Yes, the reviewer is partly correct.

On one hand, the natural source of NO is from lighting stroke, while the anthropogenic sources can be from human activities involving high temperatures, like combustion of fossil fuels.

On the other hand, the monitoring site in this study where we collected the data, Hyytiälä, is located in a boreal forest, with minor anthropogenic emissions (Heikkinen et al., 2019). Nearby sources are two saw mills and a pellet factory 6-7 km away (Äijälä et al., 2019), with no significant emission of NO. The dominant influence of air pollution are still coming from transport from industrialized areas over southern Finland, Russia and the Baltic countries (Riuttanen et al., 2013;Heikkinen et al., 2019). Primary emissions of NO from fossil combustion processes are not long-lived enough to be transported long distances. NO can react with $O_3$ rapidly to form $NO_2$, typically on the timescale of minutes. As a result, even though we cannot totally rule out the primary emission of NO during daytime, the photochemical reactions will play the dominant role in NO production. To make the statement more rigorous, revision were made in Line 388:

"During the day, photochemical reactions as well as potential emissions increase the concentration of NO, which serves as peroxy radical ($RO_2$) terminator and often outcompetes RO2 cross reactions in which dimers can be formed (Ehn et al., 2014)",

8. (Line 435), the termination of one peroxy radicals with another does not necessarily have to lead to the formation of dimers.

***Response 2.8***: As we responded in *Response 1.5,* we agreed with the reviewer that termination of two peroxy radicals doesn't necessarily lead to dimer formation. To clarify the statement, we made revision to the sentence in Line 462-463 to

"This termination step is mutually exclusive with the termination of $RO_2$ with other $RO_2$, which can lead to dimer formation."

9. (Line 497-499), rephrase the sentence.

***Response 2.9:*** The sentence has been removed from the text, to decrease the discussion part of contamination factor.

10. (Session 4.2.1) Although there might not be a daytime factor previously, it is not surprising that dimers are formed in the day. NO channel competes with the self reactions of peroxy radicals but which level of NO will really dominate is still an open question.

**Response 2.10:** We agree with the reviewer that dimers can be expected to form during daytime, however, the relative studies or results are quite few. For chamber studies, experimental results show that dimer concentration is strongly affected by NO concentration (Ehn et al., 2014). The amount of NO required to hinder dimer formation is also a function of the $RO_2$ concentrations. In ambient measurement, Mohr et al. (2017) reported a clear diel pattern of dimers with maximum after noon, which is among the first daytime gas-phase dimer observations in the atmosphere, in contrast to the typical daytime minima observed in Hyytiälä (Yan et al., 2016). The NO channel suppresses $RO_2 + RO_2$ reactions in daytime to a large extent, which leads to lower signals of these daytime dimers to be detected. But definitely, more evidence and studies are needed to reveal and quantify the NO competing ability towards dimer formation in the daytime.

---

## Author Response (AR2)

**Responses to Editor's Comments**

Comments to the Author:

One of the major issues with this paper, as identified by both reviewers, is that it focuses too much on the technical advancements and not enough on the atmospheric science implications. While the modifications have improved the manuscript somewhat, it still does not quite fall within scope because while new molecules and potential mechanisms are identified, very little attention is paid to what the implications for wider atmospheric science are (it is mentioned, but is currently buried within the text), so as such, this manuscript still comes across as a technical study where the atmospheric science is nothing more than a byproduct. I would surmise that readers unfamiliar with the fundamental mechanisms are unlikely to appreciate the significance of this work.

This should be fairly easy to remedy through the inclusion of more text in the abstract, discussion and conclusions that summarises what the general implications of these new observations are, directed at a non-specialist atmospheric science audience. Do these new molecules/mechanisms identified warrant further study, and if so, why? Could the inclusion of these mechanisms hypothetically improve NPF/SOA model performance? I would also try to better highlight in the abstract and introduction how our knowledge in this area may be incomplete and why developments like this may help to address it.

*Response to editor*

We would like to thank the editor for suggestions which helped us to improve the manuscript. Based on the editor's comments, we have revised the abstract, introduction as well as conclusions, in order to better highlight our new findings and their potential contribution to the atmospheric community, and simplify the technical part.

*The ABSTRACT* is revised now from

"With the recent developments in mass spectrometry, combined with the strengths of factor analysis techniques, our understanding of atmospheric oxidation chemistry has improved significantly. The typical approach for using techniques like positive matrix factorization (PMF) is to input all measured data for the factorization in order to separate contributions from different sources and/or processes to the total measured signal. However, while this is a valid approach for assigning the total signal to factors, we have identified several cases where useful information can be lost if solely using this approach. For example, gaseous molecules emitted from the same source can show different temporal behaviors due differing loss terms, like condensation at different rates due to different molecular masses. This conflicts with one of PMF's basic assumptions of constant factor profiles. In addition, some ranges of a mass spectrum may contain useful information, despite contributing only minimal fraction to the total signal, in which case they are unlikely to have a significant impact on the factorization result. Finally, certain mass ranges may contain molecules formed via pathways not available to molecules in other mass ranges, e.g. dimeric species versus monomeric species. In this study, we attempted to address these challenges by dividing mass spectra into sub-ranges and applying the newly developed binPMF method to these ranges separately. We utilized a dataset from a chemical ionization atmospheric pressure interface time-of-flight (CI-APi-TOF) mass spectrometer as an example. We compare the results from these three different ranges, each corresponding to molecules of different volatilities, with binPMF results from the combined range. Separate analysis showed clear benefits in dividing factors for molecules of different volatilities more accurately, in resolving different chemical processes from different ranges, and in giving a chance for high-molecular-weight molecules with low signal intensities to be used to distinguish dimeric species with different formation pathways. As two major insights from our study, we identified daytime dimer formation (diurnal peak around noon) which may contribute to NPF in Hyytiälä, as well as dimers from NO3 oxidation process. We recommend PMF users to try running their analyses on selected sub-ranges in order to further explore their datasets."

to:

"Our understanding of atmospheric oxidation chemistry has improved significantly in recent years, greatly facilitated by developments in mass spectrometry. The generated mass spectra typically contain vast amounts of information on atmospheric sources and processes, but the identification and quantification of these is hampered by the wealth of data to analyze. The implementation of factor analysis techniques have greatly facilitated this analysis, yet many atmospheric processes still remain poorly understood. Here, we present new insights on highly oxygenated products from monoterpene oxidation, measured by chemical ionization mass spectrometry, at a boreal forest site in Finland in fall 2016. Our primary focus was on the formation of accretion products, i.e. "dimers". We identified the formation of daytime dimers, with a diurnal peak at noon time, despite high NO concentrations typically expected to inhibit dimer formation. These dimers may play an important role in new particle formation events that are often observed in the forest. In addition, dimers identified as combined products of $NO_3$ and $O_3$ oxidation of monoterpenes were also found to be a large source of low-volatile vapors at night. This highlights the complexity of atmospheric oxidation chemistry, and the need for future laboratory studies on multi-oxidant systems. Neither of these two processes could have been separated without the new analysis approach deployed in our study, where we applied binned positive matrix factorization (binPMF) on sub-ranges of the mass spectra, rather than the traditional approach where the entire mass spectrum is included for PMF analysis. In addition to the main findings listed above, several other benefits compared to traditional methods were found."

In *INTRODUCTION* part, we add two paragraphs to highlight the importance of our observations and delete three paragraphs which describes the reason for and advantages of sub-range analysis. Some paragraphs were adjusted. The new introduction is as follows:

[revised manuscript text omitted]